# Debiasing Distributed Second Order Optimization with Surrogate Sketching and Scaled Regularization

**Michał Dereziński**
Department of Statistics
University of California, Berkeley
mderezin@berkeley.edu

**Burak Bartan**
Department of Electrical Engineering
Stanford University
bbartan@stanford.edu

**Mert Pilanci**
Department of Electrical Engineering
Stanford University
pilanci@stanford.edu

**Michael W. Mahoney**
ICSI and Department of Statistics
University of California, Berkeley
mmahoney@stat.berkeley.edu

## Abstract

In distributed second order optimization, a standard strategy is to average many local estimates, each of which is based on a small sketch or batch of the data. However, the local estimates on each machine are typically biased, relative to the full solution on all of the data, and this can limit the effectiveness of averaging. Here, we introduce a new technique for debiasing the local estimates, which leads to both theoretical and empirical improvements in the convergence rate of distributed second order methods. Our technique has two novel components: (1) modifying standard sketching techniques to obtain what we call a surrogate sketch; and (2) carefully scaling the global regularization parameter for local computations. Our surrogate sketches are based on determinantal point processes, a family of distributions for which the bias of an estimate of the inverse Hessian can be computed exactly. Based on this computation, we show that when the objective being minimized is $l_2$-regularized with parameter $\lambda$ and individual machines are each given a sketch of size $m$, then to eliminate the bias, local estimates should be computed using a shrunk regularization parameter given by $\lambda' = \lambda \cdot (1 - \frac{d_\lambda}{m})$, where $d_\lambda$ is the $\lambda$-effective dimension of the Hessian (or, for quadratic problems, the data matrix).

## 1 Introduction

We consider the task of second order optimization in a distributed or parallel setting. Suppose that $q$ workers are each given a small sketch of the data (e.g., a random sample or a random projection) and a parameter vector $\mathbf{x}_t$. The goal of the $k$-th worker is to construct a local estimate $\Delta_{t,k}$ of the Newton step relative to a convex loss on the full dataset. The estimates are then averaged and the parameter vector is updated using this averaged step, obtaining $\mathbf{x}_{t+1} = \mathbf{x}_t + \frac{1}{q} \sum_{k=1}^{q} \Delta_{t,k}$. This basic strategy has been extensively studied and it has proven effective for a variety of optimization tasks because of its communication-efficiency [40]. However, a key problem that limits the scalability of this approach is that local estimates of second order steps are typically biased, which means that for a sufficiently large $q$, adding more workers will not lead to any improvement in the convergence rate. Furthermore, for most types of sketched estimates this bias is difficult to compute, or even approximate, which makes it difficult to correct.

In this paper, we propose a new class of sketching methods, called *surrogate sketches*, which allow us to debias local estimates of the Newton step, thereby making distributed second order optimization more scalable. In our analysis of the surrogate sketches, we exploit recent developments in determinantal point processes (DPPs, [16]) to give exact formulas for the bias of the estimates produced with those sketches, enabling us to correct that bias. Due to algorithmic advances in DPP sampling, surrogate sketches can be implemented in time nearly linear in the size of the data, when the number of data points is much larger than their dimensionality, so our results lead to direct improvements in the time complexity of distributed second order optimization. Remarkably, our analysis of the bias of surrogate sketches leads to a simple technique for debiasing the local Newton estimates for $l_2$-regularized problems, which we call *scaled regularization*. We show that the regularizer used on the sketched data should be scaled down compared to the global regularizer, and we give an explicit formula for that scaling. Our empirical results demonstrate that scaled regularization significantly reduces the bias of local Newton estimates not only for surrogate sketches, but also for a range of other sketching techniques.

## 1.1 Debiasing via Surrogate Sketches and Scaled Regularization

Our scaled regularization technique applies to sketching the Newton step over a convex loss, as described in Section 3, however, for concreteness, we describe it here in the context of regularized least squares. Suppose that the data is given in the form of an $n \times d$ matrix $\mathbf{A}$ and an $n$-dimensional vector $\mathbf{b}$, where $n \gg d$. For a given regularization parameter $\lambda > 0$, our goal is to approximately solve the following problem:

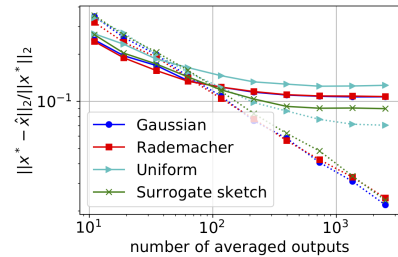

$$\mathbf{x}^* = \underset{\mathbf{x}}{\operatorname{argmin}} \ \frac{1}{2}\|\mathbf{A}\mathbf{x} - \mathbf{b}\|^2 + \frac{\lambda}{2}\|\mathbf{x}\|^2. \qquad (1)$$

Following the classical sketch-and-solve paradigm, we use a random $m \times n$ sketching matrix $\mathbf{S}$, where $m \ll n$, to replace this large $n \times d$ regularized least squares problem $(\mathbf{A}, \mathbf{b}, \lambda)$ with a smaller $m \times d$ problem of the same form. We do this by sketching both the matrix $\mathbf{A}$ and the vector $\mathbf{b}$, obtaining the problem $(\mathbf{S}\mathbf{A}, \mathbf{S}\mathbf{b}, \lambda')$ given by:

Figure 1: Estimation error against the number of averaged outputs for the Boston housing prices dataset (see Section 5). The dotted curves show the error when the regularization parameter is rescaled as in Theorem 1.

$$\hat{\mathbf{x}} = \underset{\mathbf{x}}{\operatorname{argmin}} \ \frac{1}{2}\|\mathbf{S}\mathbf{A}\mathbf{x} - \mathbf{S}\mathbf{b}\|^2 + \frac{\lambda'}{2}\|\mathbf{x}\|^2, \qquad (2)$$

where we deliberately allow $\lambda'$ to be different than $\lambda$. The question we pose is: What is the right choice of $\lambda'$ so as to minimize $\|\mathbb{E}[\hat{\mathbf{x}}] - \mathbf{x}^*\|$, i.e., the bias of $\hat{\mathbf{x}}$, which will dominate the estimation error in the case of massively parallel averaging? We show that the choice of $\lambda'$ is controlled by a classical notion of effective dimension for regularized least squares [1].

**Definition 1** *Given a matrix* $\mathbf{A}$ *and regularization parameter* $\lambda \geq 0$*, the* $\lambda$*-effective dimension of* $\mathbf{A}$ *is defined as* $d_\lambda = d_\lambda(\mathbf{A}) = \operatorname{tr}(\mathbf{A}^\top \mathbf{A}(\mathbf{A}^\top \mathbf{A} + \lambda\mathbf{I})^{-1})$.

For surrogate sketches, which we define in Section 2, it is in fact possible to bring the bias down to zero and we give an exact formula for the correct $\lambda'$ that achieves this (see Theorem 6 in Section 3 for a statement which applies more generally to the Newton's method).

**Theorem 1** *If* $\hat{\mathbf{x}}$ *is constructed using a size* $m$ *surrogate sketch from Definition 3, then:*

$$\mathbb{E}[\hat{\mathbf{x}}] = \mathbf{x}^* \quad \text{for} \quad \lambda' = \lambda \cdot \left(1 - \frac{d_\lambda}{m}\right).$$

Thus, the regularization parameter used to compute the local estimates should be smaller than the global regularizer $\lambda$. While somewhat surprising, this observation does align with some prior empirical [39] and theoretical [14] results which suggest that random sketching or sampling introduces some amount of *implicit regularization*. From this point of view, it makes sense that we should compensate for this implicit effect by reducing the amount of *explicit* regularization being used.

| | Sketch | Averaging | Regularizer | Convergence Rate | Assumption |
|---|---|---|---|---|---|
| [40] | i.i.d. row sample | uniform | $\lambda$ | $\left(\frac{1}{\alpha q} + \frac{1}{\alpha^2}\right)^t$ | $\alpha \geq 1$ |
| [15] | i.i.d. row sample | determinantal | $\lambda$ | $\left(\frac{d}{\alpha q}\right)^t$ | $\alpha \geq d$ |
| Thm. 2 | surrogate sketch | uniform | $\lambda \cdot \left(1 - \frac{d_\lambda}{m}\right)$ | $\left(\frac{1}{\alpha q}\right)^t$ | $\alpha \geq d$ |

Table 1: Comparison of convergence guarantees for the Distributed Iterative Hessian Sketch on regularized least squares (see Theorem 2), with $q$ workers and sketch size $m = \tilde{O}(\alpha\, d)$. Note that both the references [40, 15] state their results for uniform sampling sketches. This can be easily adapted to leverage score sampling, in which case each sketch costs $\tilde{O}(\mathrm{nnz}(\mathbf{A}) + \alpha d^3)$ to construct.

One might assume that the above formula for $\lambda'$ is a unique property of surrogate sketches. However, we empirically show that our scaled regularization applies much more broadly, by testing it with the standard Gaussian sketch ($\mathbf{S}$ has i.i.d. entries $\mathcal{N}(0, 1/m)$), a Rademacher sketch ($\mathbf{S}$ has i.i.d. entries equal $\frac{1}{\sqrt{m}}$ or $-\frac{1}{\sqrt{m}}$ with probability 0.5), and uniform row sampling. In Figure 1, we plot normalized estimates of the bias, $\|(\frac{1}{q}\sum_{k=1}^{q}\hat{\mathbf{x}}_k) - \mathbf{x}^*\|/\|\mathbf{x}^*\|$, by averaging $q$ i.i.d. copies of $\hat{\mathbf{x}}$, as $q$ grows to infinity, showing the results with both scaled (dotted curves) and un-scaled (solid curves) regularization. Remarkably, the scaled regularization seems to correct the bias of $\hat{\mathbf{x}}$ very effectively for Gaussian and Rademacher sketches as well as for the surrogate sketch, resulting in the estimation error decaying to zero as $q$ grows. For uniform sampling, scaled regularization also noticeably reduces the bias. In Section 5 we present experiments on more datasets which further verify these claims.

## 1.2 Convergence Guarantees for Distributed Newton Method

We use the debiasing technique introduced in Section 1.1 to obtain the main technical result of this paper, which gives a convergence and time complexity guarantee for distributed Newton's method with surrogate sketching. Once again, for concreteness, we present the result here for the regularized least squares problem (1), but a general version for convex losses is given in Section 4 (see Theorem 10). Our goal is to perform a distributed and sketched version of the classical Newton step: $\mathbf{x}_{t+1} = \mathbf{x}_t - \mathbf{H}^{-1}\mathbf{g}(\mathbf{x}_t)$, where $\mathbf{H} = \mathbf{A}^\top\mathbf{A} + \lambda\mathbf{I}$ is the Hessian of the quadratic loss, and $\mathbf{g}(\mathbf{x}_t) = \mathbf{A}^\top(\mathbf{A}\mathbf{x}_t - \mathbf{b}) + \lambda\mathbf{x}_t$ is the gradient. To efficiently approximate this step, while avoiding the $O(nd^2)$ cost of computing the exact Hessian, we use a distributed version of the so-called Iterative Hessian Sketch (IHS), which replaces the Hessian with a sketched version $\widehat{\mathbf{H}}$, but keeps the exact gradient, resulting in the update direction $\widehat{\mathbf{H}}^{-1}\mathbf{g}(\mathbf{x}_t)$ [33, 30, 27, 25]. Our goal is that $\widehat{\mathbf{H}}$ should be cheap to construct and it should lead to an unbiased estimate of the exact Newton step $\mathbf{H}^{-1}\mathbf{g}$. When the matrix $\mathbf{A}$ is sparse, it is desirable for the algorithm to run in time that depends on the input sparsity, i.e., the number of non-zeros denoted $\mathrm{nnz}(\mathbf{A})$.

**Theorem 2** *Let $\kappa$ denote the condition number of the Hessian $\mathbf{H}$, let $\mathbf{x}_0$ be the initial parameter vector and take any $\alpha \geq d$. There is an algorithm which returns a Hessian sketch $\widehat{\mathbf{H}}$ in time $O(\mathrm{nnz}(\mathbf{A})\log(n) + \alpha d^3\,\mathrm{polylog}(n, \kappa, 1/\delta))$, such that if $\widehat{\mathbf{H}}_1, ..., \widehat{\mathbf{H}}_q$ are i.i.d. copies of $\widehat{\mathbf{H}}$ then,*

$$\text{Distributed IHS:} \qquad \mathbf{x}_{t+1} = \mathbf{x}_t - \frac{1}{q}\sum_{k=1}^{q}\widehat{\mathbf{H}}_k^{-1}\mathbf{g}(\mathbf{x}_t),$$

*with probability $1 - \delta$ enjoys a linear convergence rate given as follows:*

$$\|\mathbf{x}_t - \mathbf{x}^*\|^2 \leq \rho^t \kappa \|\mathbf{x}_0 - \mathbf{x}^*\|^2, \qquad \text{where} \quad \rho = \frac{1}{\alpha q}.$$

**Remark 3** *To reach $\|\mathbf{x}_t - \mathbf{x}^*\|^2 \leq \epsilon \cdot \|\mathbf{x}_0 - \mathbf{x}^*\|^2$ we need $t \leq \frac{\log(\kappa/\epsilon)}{\log(\alpha q)}$ iterations. Also, note that the input sparsity time algorithm does not require any knowledge of $d_\lambda$ (see Appendix B for a discussion). Furthermore, the $O(\mathrm{nnz}(\mathbf{A})\log(n))$ preprocessing cost can be avoided under the assumption of low matrix coherence of the Hessian (see Definition 1 in [40]), which is often the*

*case in practice. See Theorem 10 in Section 4 for a general result on convex losses of the form $f(\mathbf{x}) = \frac{1}{n}\sum_{i=1}^{n}\ell_i(\mathbf{x}^\top\varphi_i) + \frac{\lambda}{2}\|\mathbf{x}\|^2$.*

Crucially, the linear convergence rate $\rho$ decays to zero as $q$ goes to infinity, which is possible because the local estimates of the Newton step produced by the surrogate sketch are unbiased. Just like commonly used sketching techniques, our surrogate sketch can be interpreted as replacing the matrix $\mathbf{A}$ with a smaller matrix $\mathbf{SA}$, where $\mathbf{S}$ is a $m \times n$ sketching matrix, with $m = \tilde{O}(\alpha d)$ denoting the sketch size. Unlike the Gaussian and Rademacher sketches, the sketch we use is very sparse, since it is designed to only sample and rescale a subset of rows from $\mathbf{A}$, which makes the multiplication very fast. Our surrogate sketch has two components: (1) standard i.i.d. row sampling according to the so-called $\lambda$-ridge leverage scores [20, 1]; and (2) non-i.i.d. row sampling according to a determinantal point process (DPP) [22]. While leverage score sampling has been used extensively as a sketching technique for second order methods, it typically leads to biased estimates, so combining it with a DPP is crucial to obtain strong convergence guarantees in the distributed setting. The primary computational costs in constructing the sketch come from estimating the leverage scores and sampling from the DPP.

## 1.3 Related Work

While there is extensive literature on distributed second order methods, it is useful to first compare to the most directly related approaches. In Table 1, we contrast Theorem 2 with two other results which also analyze variants of the Distributed IHS, with all sketch sizes fixed to $m = \tilde{O}(\alpha d)$. The algorithm of [40] simply uses an i.i.d. row sampling sketch to approximate the Hessian, and then uniformly averages the estimates. This leads to a bias term $\frac{1}{\alpha^2}$ in the convergence rate, which can only be reduced by increasing the sketch size. In [15], this is avoided by performing weighted averaging, instead of uniform, so that the rate decays to zero with increasing $q$. Similarly as in our work, determinants play a crucial role in correcting the bias, however with significantly different trade-offs. While they avoid having to alter the sketching method, the weighted average introduces a significant amount of variance, which manifests itself through the additional factor $d$ in the term $\frac{d}{\alpha q}$. Our surrogate sketch avoids the additional variance factor while maintaining the scalability in $q$. The only trade-off is that the time complexity of the surrogate sketch has a slightly worse polynomial dependence on $d$, and as a result we require the sketch size to be at least $\tilde{O}(d^2)$, i.e., that $\alpha \geq d$. Finally, unlike the other approaches, our method uses a scaled regularization parameter to debias the Newton estimates.

Distributed second order optimization has been considered by many other works in the literature and many methods have been proposed such as DANE [36], AIDE [35], DiSCO [42], and others [29, 2]. Distributed averaging has been discussed in the context of linear regression problems in works such as [4] and studied for ridge regression in [39]. However, unlike our approach, all of these methods suffer from biased local estimates for regularized problems. Our work deals with distributed versions of iterative Hessian sketch and Newton sketch and convergence guarantees for non-distributed version are given in [33] and [34]. Sketching for constrained and regularized convex programs and minimax optimality has been studied in [32, 41, 37]. Optimal iterative sketching algorithms for least squares problems were investigated in [24, 27, 25, 26, 28]. Bias in distributed averaging has been recently considered in [3], which provides expressions for regularization parameters for Gaussian sketches. The theoretical analysis of [3] assumes identical singular values for the data matrix whereas our results make no such assumption.

Our analysis of surrogate sketches builds upon a recent line of works which derive expectation formulas for Determinantal Point Processes (for a review, see [16]) in the context of least squares regression [17, 18, 12]. Finally, the connection we observe between the behavior of DPP sampling and Gaussian sketches has been recently demonstrated in other contexts, including generalization error of minimum-norm linear models [14] and low-rank approximation [13].

## 2 Surrogate Sketches

In this section, to motivate our surrogate sketches, we consider several standard sketching techniques and discuss their shortcomings. Our purpose in introducing surrogate sketches is to enable exact

analysis of the sketching bias in second order optimization, thereby permitting us to find the optimal hyper-parameters for distributed averaging.

Given an $n \times d$ data matrix $\mathbf{A}$, we define a standard sketch of $\mathbf{A}$ as the matrix $\mathbf{SA}$, where $\mathbf{S} \sim \mathcal{S}_\mu^m$ is a random $m \times n$ matrix with $m$ i.i.d. rows distributed according to measure $\mu$ with identity covariance, rescaled so that $\mathbb{E}[\mathbf{S}^\top \mathbf{S}] = \mathbf{I}$. This includes such standard sketches as:

1. *Gaussian sketch:* each row of $\mathbf{S}$ is distributed as $\mathcal{N}(\mathbf{0}, \frac{1}{m}\mathbf{I})$.

2. *Rademacher sketch:* each entry of $\mathbf{S}$ is $\frac{1}{\sqrt{m}}$ with probability $1/2$ and $-\frac{1}{\sqrt{m}}$ otherwise.

3. *Row sampling:* each row of $\mathbf{S}$ is $\frac{1}{\sqrt{p_i m}}\mathbf{e}_i$, where $\Pr\{i\} = p_i$ and $\sum_i p_i = 1$.

Here, the row sampling sketch can be uniform (which is common in practice), and it also includes row norm squared sampling and leverage score sampling (which leads to better results), where the distribution $p_i$ depends on the data matrix $\mathbf{A}$.

Standard sketches are generally chosen so that the sketched covariance matrix $\mathbf{A}^\top \mathbf{S}^\top \mathbf{SA}$ is an unbiased estimator of the full data covariance matrix, $\mathbf{A}^\top \mathbf{A}$. This is ensured by the fact that $\mathbb{E}[\mathbf{S}^\top \mathbf{S}] = \mathbf{I}$. However, in certain applications, it is not the data covariance matrix itself that is of primary interest, but rather its inverse. In this case, standard sketching techniques no longer yield unbiased estimators. Our surrogate sketches aim to correct this bias, so that, for example, we can construct an unbiased estimator for the regularized inverse covariance matrix, $(\mathbf{A}^\top \mathbf{A} + \lambda \mathbf{I})^{-1}$ (given some $\lambda > 0$). This is important for regularized least squares and second order optimization.

We now give the definition of a surrogate sketch. Consider some $n$-variate measure $\mu$, and let $\mathbf{X} \sim \mu^m$ be the i.i.d. random design of size $m$ for $\mu$, i.e., an $m \times n$ random matrix with i.i.d. rows drawn from $\mu$. Without loss of generality, assume that $\mu$ has identity covariance, so that $\mathbb{E}[\mathbf{X}^\top \mathbf{X}] = m\mathbf{I}$. In particular, this implies that $\frac{1}{\sqrt{m}}\mathbf{X} \sim \mathcal{S}_\mu^m$ is a random sketching matrix.

Before we introduce the surrogate sketch, we define a so-called *determinantal design* (an extension of the definitions proposed by [14, 18]), which uses determinantal rescaling to transform the distribution of $\mathbf{X}$ into a non-i.i.d. random matrix $\bar{\mathbf{X}}$. The transformation is parameterized by the matrix $\mathbf{A}$, the regularization parameter $\lambda > 0$ and a parameter $\gamma > 0$ which controls the size of the matrix $\bar{\mathbf{X}}$.

**Definition 2** *Given scalars $\lambda, \gamma > 0$ and a matrix $\mathbf{A} \in \mathbb{R}^{n \times d}$, we define the determinantal design $\bar{\mathbf{X}} \sim \mathrm{Det}_\mu^\gamma(\mathbf{A}, \lambda)$ as a random matrix with randomized row-size, so that*

$$\Pr\left\{\bar{\mathbf{X}} \in E\right\} \propto \mathbb{E}\Big[\det(\mathbf{A}^\top \mathbf{X}^\top \mathbf{X} \mathbf{A} + \lambda\gamma\mathbf{I}) \cdot \mathbf{1}_{[\mathbf{X} \in E]}\Big], \quad where \quad \mathbf{X} \sim \mu^M, \quad M \sim \mathrm{Poisson}(\gamma).$$

We next give the key properties of determinantal designs that make them useful for sketching and second-order optimization. The following lemma is an extension of the results shown for determinantal point processes by [14].

**Lemma 4** *Let $\bar{\mathbf{X}} \sim \mathrm{Det}_\mu^\gamma(\mathbf{A}, \lambda)$. Then, we have:*

$$\mathbb{E}\Big[\big(\mathbf{A}^\top \bar{\mathbf{X}}^\top \bar{\mathbf{X}} \mathbf{A} + \lambda\gamma\mathbf{I}\big)^{-1} \mathbf{A}^\top \bar{\mathbf{X}}^\top \bar{\mathbf{X}}\Big] = \big(\mathbf{A}^\top \mathbf{A} + \lambda\mathbf{I}\big)^{-1} \mathbf{A}^\top,$$

$$\mathbb{E}\Big[\big(\mathbf{A}^\top \bar{\mathbf{X}}^\top \bar{\mathbf{X}} \mathbf{A} + \lambda\gamma\mathbf{I}\big)^{-1}\Big] = \gamma^{-1}\big(\mathbf{A}^\top \mathbf{A} + \lambda\mathbf{I}\big)^{-1}.$$

The row-size of $\bar{\mathbf{X}}$, denoted by $\#(\bar{\mathbf{X}})$, is a random variable, and this variable is *not* distributed according to $\mathrm{Poisson}(\gamma)$, even though $\gamma$ can be used to control its expectation. As a result of the determinantal rescaling, the distribution of $\#(\bar{\mathbf{X}})$ is shifted towards larger values relative to $\mathrm{Poisson}(\gamma)$, so that its expectation becomes:

$$\mathbb{E}\big[\#(\bar{\mathbf{X}})\big] = \gamma + d_\lambda, \quad where \quad d_\lambda = \mathrm{tr}(\mathbf{A}^\top \mathbf{A}(\mathbf{A}^\top \mathbf{A} + \lambda\mathbf{I})^{-1}).$$

We can now define the *surrogate sketching matrix* $\bar{\mathbf{S}}$ by rescaling the matrix $\bar{\mathbf{X}}$, similarly to how we defined the standard sketching matrix $\mathbf{S} = \frac{1}{\sqrt{m}}\mathbf{X}$ for $\mathbf{X} \sim \mu^m$.

**Definition 3** *Let $m > d_\lambda$. Moreover, let $\gamma > 0$ be the unique positive scalar for which $\mathbb{E}[\#(\bar{\mathbf{X}})] = m$, where $\bar{\mathbf{X}} \sim \mathrm{Det}_\mu^\gamma(\mathbf{A}, \lambda)$. Then, $\bar{\mathbf{S}} = \frac{1}{\sqrt{m}}\bar{\mathbf{X}} \sim \bar{\mathcal{S}}_\mu^m(\mathbf{A}, \lambda)$ is a surrogate sketching matrix for $\mathcal{S}_\mu^m$.*

Note that many different surrogate sketches can be defined for a single sketching distribution $\mathcal{S}_\mu^m$, depending on the choice of $\mathbf{A}$ and $\lambda$. In particular, this means that a surrogate sketching distribution (even when the pre-surrogate i.i.d. distribution is Gaussian or the uniform distribution) always depends on the data matrix $\mathbf{A}$, whereas many standard sketches (such as Gaussian and uniform) are oblivious to the data matrix.

Of particular interest to us is the class of surrogate *row sampling* sketches, i.e. where the probability measure $\mu$ is defined by $\mu\big(\{\frac{1}{\sqrt{p_i}}\mathbf{e}_i^\top\}\big) = p_i$ for $\sum_{i=1}^n p_i = 1$. In this case, we can straightforwardly leverage the algorithmic results on sampling from determinantal point processes [10, 11] to obtain efficient algorithms for constructing surrogate sketches.

**Theorem 5** *Given any $n \times d$ matrix $\mathbf{A}$, $\lambda > 0$ and $(p_1, ..., p_n)$, we can construct the surrogate row sampling sketch with respect to $p$ (of any size $m \leq n$) in time $O(\mathrm{nnz}(\mathbf{A})\log(n) + d^4\log(d))$.*

## 3 Unbiased Estimates for the Newton Step

Consider a convex minimization problem defined by the following loss function:

$$f(\mathbf{x}) = \frac{1}{n}\sum_{i=1}^n \ell_i(\mathbf{x}^\top\varphi_i) + \frac{\lambda}{2}\|\mathbf{x}\|^2,$$

where each $\ell_i$ is a twice differentiable convex function and $\varphi_1, ..., \varphi_n$ are the input feature vectors in $\mathbb{R}^d$. For example, if $\ell_i(z) = \frac{1}{2}(z - b_j)^2$, then we recover the regularized least squares task; and if $\ell_i(z) = \log(1 + \mathrm{e}^{-zb_j})$, then we recover logistic regression. The Newton's update for this minimization task can be written as follows:

$$\mathbf{x}_{t+1} = \mathbf{x}_t - \overbrace{\Big(\frac{1}{n}\sum_i \ell_i''(\mathbf{x}_t^\top\varphi_i)\,\varphi_i\varphi_i^\top + \lambda\mathbf{I}\Big)}^{\text{Hessian } \mathbf{H}(\mathbf{x}_t)}{}^{-1}\overbrace{\Big(\frac{1}{n}\sum_i \ell_i'(\mathbf{x}^\top\varphi_i)\,\varphi_i\ +\ \lambda\mathbf{x}_t\Big)}^{\text{gradient } \mathbf{g}(\mathbf{x}_t)}.$$

Newton's method can be interpreted as solving a regularized least squares problem which is the local approximation of $f$ at the current iterate $\mathbf{x}_t$. Thus, with the appropriate choice of matrix $\mathbf{A}_t$ (consisting of scaled row vectors $\varphi_i^\top$) and vector $\mathbf{b}_t$, the Hessian and gradient can be written as: $\mathbf{H}(\mathbf{x}_t) = \mathbf{A}_t^\top\mathbf{A}_t + \lambda\mathbf{I}$ and $\mathbf{g}(\mathbf{x}_t) = \mathbf{A}_t^\top\mathbf{b}_t + \lambda\mathbf{x}_t$. We now consider two general strategies for sketching the Newton step, both of which we discussed in Section 1 for regularized least squares.

### 3.1 Sketch-and-Solve

We first analyze the classic sketch-and-solve paradigm which has been popularized in the context of least squares, but also applies directly to the Newton's method. This approach involves constructing sketched versions of both the Hessian and the gradient, by sketching with a random matrix $\mathbf{S}$. Crucially, we modify this classic technique by allowing the regularization parameter to be different than in the global problem, obtaining the following sketched version of the Newton step:

$$\hat{\mathbf{x}}_{\mathrm{SaS}} = \mathbf{x}_t - \widetilde{\mathbf{H}}_t^{-1}\tilde{\mathbf{g}}_t, \quad \text{for} \quad \widetilde{\mathbf{H}}_t = \mathbf{A}_t^\top\mathbf{S}^\top\mathbf{S}\mathbf{A}_t + \lambda'\mathbf{I}, \quad \tilde{\mathbf{g}}_t = \mathbf{A}_t^\top\mathbf{S}^\top\mathbf{S}\mathbf{b}_t + \lambda'\mathbf{x}_t.$$

Our goal is to obtain an unbiased estimate of the full Newton step, i.e., such that $\mathbb{E}[\hat{\mathbf{x}}_{\mathrm{SaS}}] = \mathbf{x}_{t+1}$, by combining a surrogate sketch with an appropriately scaled regularization $\lambda'$.

We now establish the correct choice of surrogate sketch and scaled regularization to achieve unbiasedness. The following result is a more formal and generalized version of Theorem 1. We let $\mu$ be any distribution that satisfies the assumptions of Definition 3, so that $\mathcal{S}_\mu^m$ corresponds to any one of the standard sketches discussed in Section 2.

**Theorem 6** *If $\hat{\mathbf{x}}_{\mathrm{SaS}}$ is constructed using a surrogate sketch $\mathbf{S} \sim \bar{\mathcal{S}}_\mu^m(\mathbf{A}_t, \lambda)$ of size $m > d_\lambda$, then:*

$$\mathbb{E}[\hat{\mathbf{x}}_{\mathrm{SaS}}] = \mathbf{x}_{t+1} \quad \text{for} \quad \lambda' = \lambda \cdot \Big(1 - \frac{d_\lambda}{m}\Big).$$

## 3.2 Newton Sketch

We now consider the method referred to as the Newton Sketch [34, 31], which differs from the sketch-and-solve paradigm in that it only sketches the Hessian, whereas the gradient is computed exactly. Note that in the case of least squares, this algorithm exactly reduces to the Iterative Hessian Sketch, which we discussed in Section 1.2. This approach generally leads to more accurate estimates than sketch-and-solve, however it requires exact gradient computation, which in distributed settings often involves an additional communication round. Our Newton Sketch estimate uses the same $\lambda'$ as for the sketch-and-solve, however it enters the Hessian somewhat differently:

$$\hat{\mathbf{x}}_{\text{NS}} = \mathbf{x}_t - \widehat{\mathbf{H}}_t^{-1}\mathbf{g}(\mathbf{x}_t), \quad \text{for} \quad \widehat{\mathbf{H}}_t = \tfrac{\lambda}{\lambda'}\mathbf{A}_t^\top\mathbf{S}^\top\mathbf{S}\mathbf{A}_t + \lambda\mathbf{I} = \tfrac{\lambda}{\lambda'}\widetilde{\mathbf{H}}_t.$$

The additional factor $\frac{\lambda}{\lambda'}$ comes as a result of using the exact gradient. One way to interpret it is that we are scaling the data matrix $\mathbf{A}_t$ instead of the regularization. The following result shows that, with $\lambda'$ chosen as before, the surrogate Newton Sketch is unbiased.

**Theorem 7** *If $\hat{\mathbf{x}}_{\text{NS}}$ is constructed using a surrogate sketch $\mathbf{S} \sim \bar{\mathcal{S}}_\mu^m(\mathbf{A}_t, \lambda)$ of size $m > d_\lambda$, then:*

$$\mathbb{E}[\hat{\mathbf{x}}_{\text{NS}}] = \mathbf{x}_{t+1} \quad \text{for} \quad \lambda' = \lambda \cdot \left(1 - \frac{d_\lambda}{m}\right).$$

# 4  Convergence Analysis

Here, we study the convergence guarantees of the surrogate Newton Sketch with distributed averaging. Consider $q$ i.i.d. copies $\widehat{\mathbf{H}}_{t,1}, ..., \widehat{\mathbf{H}}_{t,q}$ of the Hessian sketch $\widehat{\mathbf{H}}_t$ defined in Section 3.2. We start by finding an upper bound for the distance between the optimal Newton update and averaged Newton sketch update at the $t$'th iteration, defined as $\hat{\mathbf{x}}_{t+1} = \mathbf{x}_t - \frac{1}{q}\sum_{k=1}^q \widehat{\mathbf{H}}_{t,k}^{-1}\mathbf{g}(\mathbf{x}_t)$. We will use Mahalanobis norm as the distance metric. Let $\|\mathbf{v}\|_{\mathbf{M}}$ denote the Mahalanobis norm, i.e., $\|\mathbf{v}\|_{\mathbf{M}} = \sqrt{\mathbf{v}^\top\mathbf{M}\mathbf{v}}$. The distance between the updates is equal to the distance between the next iterates:

$$\|\mathbf{x}_{t+1} - \hat{\mathbf{x}}_{t+1}\|_{\mathbf{H}_t} = \|(\bar{\mathbf{H}}_t^{-1} - \mathbf{H}_t^{-1})\mathbf{g}_t\|_{\mathbf{H}_t}, \quad \text{where} \quad \bar{\mathbf{H}}_t^{-1} = \frac{1}{q}\sum_{k=1}^q \widehat{\mathbf{H}}_{t,k}^{-1}.$$

We can bound this quantity in terms of the spectral norm approximation error of $\bar{\mathbf{H}}_t^{-1}$ as follows:

$$\|(\bar{\mathbf{H}}_t^{-1} - \mathbf{H}_t^{-1})\mathbf{g}_t\|_{\mathbf{H}_t} \leq \|\mathbf{H}_t^{\frac{1}{2}}(\bar{\mathbf{H}}_t^{-1} - \mathbf{H}_t^{-1})\mathbf{H}_t^{\frac{1}{2}}\| \cdot \|\mathbf{H}_t^{-1}\mathbf{g}_t\|_{\mathbf{H}_t}.$$

Note that the second term, $\mathbf{H}_t^{-1}\mathbf{g}_t$, is the exact Newton step. To upper bound the first term, we now focus our discussion on a particular variant of surrogate sketch $\bar{\mathcal{S}}_\mu^m(\mathbf{A}, \lambda)$ that we call surrogate leverage score sampling. Leverage score sampling is an i.i.d. row sampling method, i.e., the probability measure $\mu$ is defined by $\mu(\{\frac{1}{\sqrt{p_i}}\mathbf{e}_i^\top\}) = p_i$ for $\sum_{i=1}^n p_i = 1$. Specifically, we consider the so-called $\lambda$-ridge leverage scores which have been used in the context of regularized least squares [1], where the probabilities must satisfy $p_i \geq \frac{1}{2}\mathbf{a}_i^\top(\mathbf{A}_t^\top\mathbf{A}_t + \lambda\mathbf{I})^{-1}\mathbf{a}_i/d_\lambda$ ($\mathbf{a}_i^\top$ denotes a row of $\mathbf{A}_t$). Such $p_i$'s can be found efficiently using standard random projection techniques [19, 9].

**Lemma 8** *If $n \geq m \geq C\alpha d_\lambda \text{polylog}(n, \kappa, 1/\delta)$ and we use the surrogate leverage score sampling sketch of size $m$, then the i.i.d. copies $\widehat{\mathbf{H}}_{t,1}, ..., \widehat{\mathbf{H}}_{t,q}$ of the sketch $\widehat{\mathbf{H}}_t$ with probability $1 - \delta$ satisfy:*

$$\|\mathbf{H}_t^{\frac{1}{2}}(\bar{\mathbf{H}}_t^{-1} - \mathbb{E}[\bar{\mathbf{H}}_t^{-1}])\mathbf{H}_t^{\frac{1}{2}}\| \leq \frac{1}{\sqrt{\alpha q}}, \quad \text{where} \quad \bar{\mathbf{H}}_t^{-1} = \frac{1}{q}\sum_{k=1}^q \widehat{\mathbf{H}}_{t,k}^{-1}.$$

Note that, crucially, we can invoke the unbiasedness of the Hessian sketch, $\mathbb{E}[\bar{\mathbf{H}}_t^{-1}] = \mathbf{H}_t^{-1}$, so we obtain that with probability at least $1 - \delta$,

$$\|\mathbf{x}_{t+1} - \hat{\mathbf{x}}_{t+1}\|_{\mathbf{H}_t} \leq \frac{1}{\sqrt{\alpha q}} \cdot \|\mathbf{H}_t^{-1}\mathbf{g}_t\|_{\mathbf{H}_t}. \tag{3}$$

We now move on to measuring how close the next Newton sketch iterate is to the global optimizer of the loss function $f(\mathbf{x})$. For this part of the analysis, we assume that the Hessian matrix is $L$-Lipschitz.

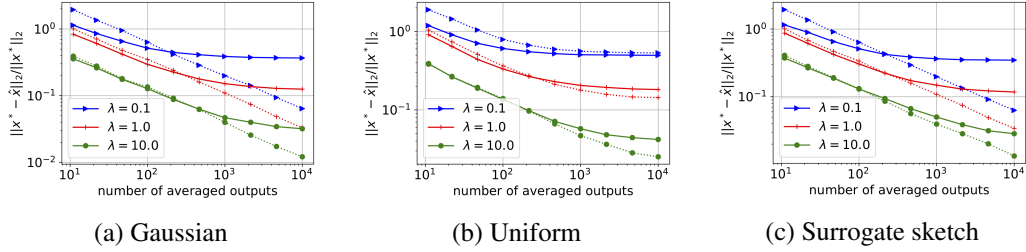

(a) Gaussian        (b) Uniform        (c) Surrogate sketch

Figure 2: Estimation error against the number of averaged outputs for regularized least squares on first two classes of Cifar-10 dataset ($n = 10000$, $d = 3072$, $m = 1000$) for different regularization parameter values $\lambda$. The dotted lines show the error for the debiased versions (obtained using $\lambda'$ expressions) for each straight line with the same color and marker.

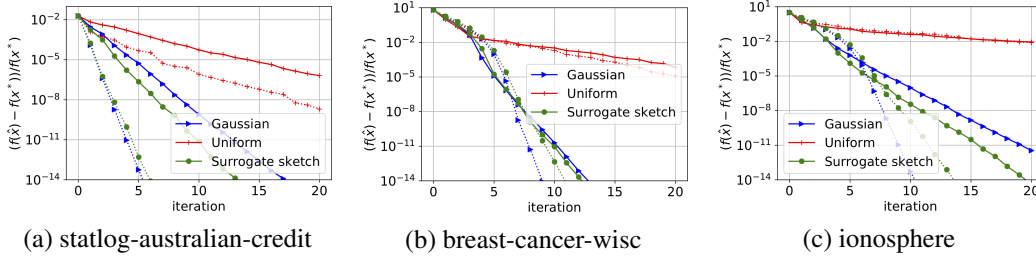

(a) statlog-australian-credit      (b) breast-cancer-wisc      (c) ionosphere

Figure 3: Distributed Newton sketch algorithm for logistic regression with $\ell_2$-regularization on different UCI datasets. The dotted curves show the error for when the regularization parameter is rescaled using the provided expression for $\lambda'$. In all the experiments, we have $q = 100$ workers and $\lambda = 10^{-4}$. The dimensions for each dataset are $(690 \times 14)$, $(699 \times 9)$, $(351 \times 33)$, and the sketch sizes are $m = 50, 50, 100$ for plots a,b,c, respectively. The step size for distributed Newton sketch updates has been determined via backtracking line search with parameters $\tau = 2$, $c = 0.1$, $a_0 = 1$.

**Assumption 9** *The Hessian matrix $\mathbf{H}(\mathbf{x})$ is $L$-Lipschitz continuous, that is, $\|\mathbf{H}(\mathbf{x}) - \mathbf{H}(\mathbf{x}')\| \leq L\|\mathbf{x} - \mathbf{x}'\|$ for all $\mathbf{x}$ and $\mathbf{x}'$.*

Combining (3) with Lemma 14 from [15] and letting $\eta = 1/\sqrt{\alpha}$, we obtain the following convergence result for the distributed Newton Sketch using surrogate leverage score sampling sketch.

**Theorem 10** *Let $\kappa$ and $\lambda_{\min}$ be the condition number and smallest eigenvalue of the Hessian $\mathbf{H}(\mathbf{x}_t)$, respectively. The distributed Newton Sketch update constructed using a surrogate leverage score sampling sketch of size $m = O(\alpha d \cdot \mathrm{polylog}(n, \kappa, 1/\delta))$ and averaged over $q$ workers, satisfies:*

$$\|\hat{\mathbf{x}}_{t+1} - \mathbf{x}^*\| \leq \max\left\{ \frac{1}{\sqrt{\alpha q}}\sqrt{\kappa}\|\mathbf{x}_t - \mathbf{x}^*\|, \frac{2L}{\lambda_{\min}}\|\mathbf{x}_t - \mathbf{x}^*\|^2 \right\}.$$

**Remark 11** *The convergence rate for the distributed Iterative Hessian Sketch algorithm as given in Theorem 2 is obtained by using (3) with $\mathbf{H}_t = \mathbf{A}^\top\mathbf{A} + \lambda\mathbf{I}$. The assumption that $\alpha \geq d$ in Theorem 2 is only needed for the time complexity (see Theorem 5). The convergence rate holds for $\alpha \geq 1$.*

## 5 Numerical Results

In this section we present numerical results, with further details provided in Appendix D. Figures 2 and 4 show the estimation error as a function of the number of averaged outputs for the regularized least squares problem discussed in Section 1.1, on Cifar-10 and Boston housing prices datasets, respectively.

Figure 2 illustrates that when the number of averaged outputs is large, rescaling the regularization parameter using the expression $\lambda' = \lambda \cdot (1 - \frac{d_\lambda}{m})$, as in Theorem 1, improves on the estimation error for a range of different $\lambda$ values. We observe that this is true not only for the surrogate sketch but also for the Gaussian sketch (we also tested the Rademacher sketch, which performed exactly as the Gaussian did). For uniform sampling, rescaling the regularization parameter does not lead to an

unbiased estimator, but it significantly reduces the bias in most instances. Figure 4 compares the surrogate row sampling sketch to the standard i.i.d. row sampling used in conjunction with averaging methods suggested by [40] (unweighted averaging) and [15] (determinantal averaging), on the Boston housing dataset. We used: $\lambda = 10$, $\lambda' = 4.06$, and sketch size $m = 50$. We show an average over 100 trials, along with the standard error. We observe that the better theoretical guarantees achieved by the surrogate sketch, as shown in Table 1, translate to improved empirical performance.

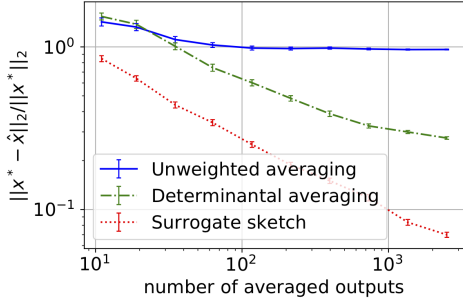

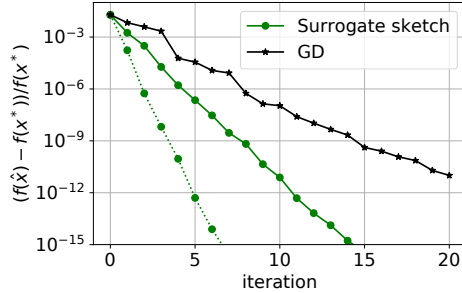

Figure 4: Estimation error of the *surrogate sketch*, against uniform sampling with *unweighted averaging* [40] and *determinantal averaging* [15].

Figure 5: Distributed Newton sketch algorithm ($m = 50$, $q = 100$) vs full gradient descent (GD). Dotted curve shows the case with scaled regularization (this does not apply to full GD).

Figure 3 shows the estimation error against iterations for the distributed Newton sketch algorithm running on a logistic regression problem with $\ell_2$ regularization on three different binary classification UCI datasets. We observe that the rescaled regularization technique leads to significant speedups in convergence, particularly for Gaussian and surrogate sketches.

Figure 5 shows an empirical comparison of the convergence rates of the distributed Newton sketch algorithm and the gradient descent algorithm for a two-class logistic regression problem with $\ell_2$ regularization solved on the UCI dataset statlog-australian-credit. We note that the gradient descent algorithm in this experiment uses exact full gradients and the learning rate is determined via backtracking line search. The results of Figure 5 demonstrate that to reach a relative error of $10^{-9}$, gradient descent requires 3 times more iterations than does distributed Newton sketch with surrogate sketching and scaled regularization. We note that it is to be expected that gradient descent requires more iterations to reach a given error compared to distributed Newton sketch, since it only utilizes first order information.

## 6   Conclusion

We introduced two techniques for debiasing distributed second order methods. First, we defined a family of sketching methods called *surrogate sketches*, which admit exact bias expressions for local Newton estimates. Second, we proposed *scaled regularization*, a method for correcting that bias.

## Broader Impact

This work does not present any foreseeable negative societal consequence. We believe that the proposed optimization methods in this work can have positive societal impacts. Our main results can be applied in massive scale distributed learning and optimization problems which are frequently encountered in practical AI systems. Using our methods, the learning phase can be significantly accelerated and consequently energy costs for training can be significantly reduced.

## Acknowledgements

This work was partially supported by the National Science Foundation under grants IIS-1838179 and ECCS-2037304, Facebook Research, Adobe Research and Stanford SystemX Alliance. Also, MD and MWM acknowledge DARPA, NSF, and ONR for providing partial support of this work.

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
