[Supplementary Material]

# A Expectation Formulas for Surrogate Sketches

In this section show the expectation formulas given in Lemma 4. First, we derive the normalization constant of the determinantal design introduced in Definition 2. For this, we rely on the framework of *determinant preserving random matrices* recently introduced by [14]. The proofs here roughly follow the techniques from [14], the main difference being that we consider regularized matrices, whereas they focus on the unregularized case.

A square random matrix is determinant preserving (d.p.) if taking expectation commutes with computing a determinant for that matrix and all its submatrices. Consider the matrix $\mathbf{X} \sim \mu^M$ for an isotropic $d$-variate measure $\mu$ and $M \sim \text{Poisson}(\gamma)$, as in Definition 2. In Lemma 5, [14] show that the matrix $\mathbf{A}^\top \mathbf{X}^\top \mathbf{X} \mathbf{A}$ is determinant preserving. Thus, using closure under addition (Lemma 4 in [14]), the matrix $\mathbf{A}^\top \mathbf{X}^\top \mathbf{X} \mathbf{A} + \lambda \gamma \mathbf{I}$ is also d.p., so the normalization constant for the probability defined in Definition 2 is:

$$\mathbb{E}\big[ \det(\mathbf{A}^\top \mathbf{X}^\top \mathbf{X} \mathbf{A} + \lambda \gamma \mathbf{I}) \big] = \det\big(\mathbf{A}^\top \mathbb{E}[\mathbf{X}^\top \mathbf{X}]\mathbf{A} + \lambda \gamma \mathbf{I}\big) = \det(\gamma \mathbf{A}^\top \mathbf{A} + \lambda \gamma \mathbf{I}).$$

**Proof of Lemma 4** By definition, any d.p. matrix $\mathbf{C}$ satisfies $\mathbb{E}[\text{adj}(\mathbf{C})] = \text{adj}(\mathbb{E}[\mathbf{C}])$, where $\text{adj}(\mathbf{C})$ denotes the adjugate of a square matrix, which for any positive definite matrix is given by $\text{adj}(\mathbf{C}) = \det(\mathbf{C})\mathbf{C}^{-1}$. This allows us to show the second expectation formula from Lemma 4 for $\bar{\mathbf{X}} \sim \text{Det}_\mu^\gamma(\mathbf{A}, \lambda)$. Note that the proof is analogous to the proof of Lemma 11 in [14].

$$\mathbb{E}\Big[ \big(\mathbf{A}^\top \bar{\mathbf{X}}^\top \bar{\mathbf{X}} \mathbf{A} + \lambda \gamma \mathbf{I}\big)^{-1} \Big] = \frac{\mathbb{E}[\det(\mathbf{A}^\top \mathbf{X}^\top \mathbf{X} \mathbf{A} + \lambda \gamma \mathbf{I}) \cdot (\mathbf{A}^\top \mathbf{X}^\top \mathbf{X} \mathbf{A} + \lambda \gamma \mathbf{I})^{-1}]}{\det(\gamma \mathbf{A}^\top \mathbf{A} + \lambda \gamma \mathbf{I})}$$

$$= \frac{\mathbb{E}[\text{adj}(\mathbf{A}^\top \mathbf{X}^\top \mathbf{X} \mathbf{A} + \lambda \gamma \mathbf{I})]}{\det(\gamma \mathbf{A}^\top \mathbf{A} + \lambda \gamma \mathbf{I})}$$

$$= \frac{\text{adj}(\gamma \mathbf{A}^\top \mathbf{A} + \lambda \gamma \mathbf{I})}{\det(\gamma \mathbf{A}^\top \mathbf{A} + \lambda \gamma \mathbf{I})} = (\gamma \mathbf{A}^\top \mathbf{A} + \lambda \gamma \mathbf{I})^{-1}.$$

We next prove the first expectation formula from Lemma 4, by following the steps outlined by [14] in the proof of their Lemma 13. Let $\mathbf{b}$ denote any vector in $\mathbb{R}^n$. The $i$th entry of the vector $\big(\mathbf{A}^\top \mathbf{X}^\top \mathbf{X} \mathbf{A} + \lambda \gamma \mathbf{I}\big)^{-1} \mathbf{A}^\top \mathbf{X}^\top \mathbf{X} \mathbf{b}$ can be obtained by left multiplying it by the $i$th standard basis vector $\mathbf{e}_i^\top$. We will use the following observation (Fact 2.14.2 from [5]):

$$\mathbf{e}_i^\top \text{adj}\big(\mathbf{A}^\top \mathbf{X}^\top \mathbf{X} \mathbf{A} + \lambda \gamma \mathbf{I}\big)^{-1} \mathbf{A}^\top \mathbf{X}^\top \mathbf{X} \mathbf{b} = \det(\mathbf{A}^\top \mathbf{X}^\top \mathbf{X} \mathbf{A} + \lambda \gamma \mathbf{I} + \mathbf{A}^\top \mathbf{X}^\top \mathbf{X} \mathbf{b} \mathbf{e}_i^\top)$$
$$- \det(\mathbf{A}^\top \mathbf{X}^\top \mathbf{X} \mathbf{A} + \lambda \gamma \mathbf{I}).$$

Combining this with the fact that both the matrices $\mathbf{A}^\top \mathbf{X}^\top \mathbf{X} \mathbf{A} + \lambda \gamma \mathbf{I}$ and $\mathbf{A}^\top \mathbf{X}^\top \mathbf{X}(\mathbf{A} + \mathbf{b} \mathbf{e}_i^\top) + \lambda \gamma \mathbf{I}$ are determinant preserving for $\mathbf{X}$ defined as before, we obtain that:

$$\mathbb{E}\big[\mathbf{e}_i^\top \big(\mathbf{A}^\top \bar{\mathbf{X}}^\top \bar{\mathbf{X}} \mathbf{A} + \lambda \gamma \mathbf{I}\big)^{-1} \mathbf{A}^\top \bar{\mathbf{X}}^\top \bar{\mathbf{X}} \mathbf{b}\big] = \frac{\mathbb{E}\big[\mathbf{e}_i^\top \text{adj}\big(\mathbf{A}^\top \mathbf{X}^\top \mathbf{X} \mathbf{A} + \lambda \gamma \mathbf{I}\big)^{-1} \mathbf{A}^\top \mathbf{X}^\top \mathbf{X} \mathbf{b}\big]}{\det(\gamma \mathbf{A}^\top \mathbf{A} + \lambda \gamma \mathbf{I})}$$

$$= \frac{\mathbb{E}[\det(\mathbf{A}^\top \mathbf{X}^\top \mathbf{X}(\mathbf{A} + \mathbf{b} \mathbf{e}_i^\top) + \lambda \gamma \mathbf{I}) - \det(\mathbf{A}^\top \mathbf{X}^\top \mathbf{X} \mathbf{A} + \lambda \gamma \mathbf{I})]}{\det(\gamma \mathbf{A}^\top \mathbf{A} + \lambda \gamma \mathbf{I})}$$

$$= \frac{\det(\gamma \mathbf{A}^\top (\mathbf{A} + \mathbf{b} \mathbf{e}_i^\top) + \lambda \gamma \mathbf{I}) - \det(\gamma \mathbf{A}^\top \mathbf{A} + \lambda \gamma \mathbf{I})}{\det(\gamma \mathbf{A}^\top \mathbf{A} + \lambda \gamma \mathbf{I})}$$

$$= \frac{\det(\gamma \mathbf{A}^\top \mathbf{A} + \lambda \gamma \mathbf{I})\mathbf{e}_i^\top (\gamma \mathbf{A}^\top \mathbf{A} + \lambda \gamma \mathbf{I})^{-1} \gamma \mathbf{A}^\top \mathbf{b}}{\det(\gamma \mathbf{A}^\top \mathbf{A} + \lambda \gamma \mathbf{I})} = \mathbf{e}_i^\top (\mathbf{A}^\top \mathbf{A} + \lambda \mathbf{I})^{-1} \mathbf{A}^\top \mathbf{b}.$$

Since the above holds for all indices $i$ and all vectors $\mathbf{b}$, this completes the proof. ∎

We next use Lemma 4 to prove Theorems 6 and 7.

**Proof of Theorem 6** Suppose that, in Lemma 4, parameter $\gamma$ is chosen as in Definition 3 and let $\bar{\mathbf{X}} \sim \text{Det}_\mu^\gamma(\mathbf{A}, \lambda)$. Then we have $m = \mathbb{E}[\#(\bar{\mathbf{X}})] = \gamma + d_\lambda(\mathbf{A})$ and the surrogate sketch in Theorem

6 is given by $\mathbf{S} = \frac{1}{\sqrt{m}}\bar{\mathbf{X}}$. Note that we have $\lambda' = \lambda \cdot \gamma/m$, so we can write:

$$
\begin{aligned}
\mathbb{E}\big[\widetilde{\mathbf{H}}_t^{-1}\tilde{\mathbf{g}}_t\big] &= \mathbb{E}\Big[\big(\mathbf{A}_t^\top\mathbf{S}^\top\mathbf{S}\mathbf{A}_t + \lambda'\mathbf{I}\big)^{-1}\big(\mathbf{A}_t^\top\mathbf{S}^\top\mathbf{S}\mathbf{b}_t + \lambda'\mathbf{x}_t\big)\Big] \\
&= \mathbb{E}\Big[\big(\tfrac{1}{m}\mathbf{A}_t^\top\bar{\mathbf{X}}^\top\bar{\mathbf{X}}\mathbf{A}_t + \lambda\tfrac{\gamma}{m}\mathbf{I}\big)^{-1}\tfrac{1}{m}\mathbf{A}_t^\top\bar{\mathbf{X}}^\top\bar{\mathbf{X}}\mathbf{b}_t\Big] + \mathbb{E}\Big[\big(\tfrac{1}{m}\mathbf{A}_t^\top\bar{\mathbf{X}}^\top\bar{\mathbf{X}}\mathbf{A}_t + \lambda\tfrac{\gamma}{m}\mathbf{I}\big)^{-1}\lambda\tfrac{\gamma}{m}\mathbf{x}_t\Big] \\
&= \mathbb{E}\Big[\big(\mathbf{A}_t^\top\bar{\mathbf{X}}^\top\bar{\mathbf{X}}\mathbf{A}_t + \lambda\gamma\mathbf{I}\big)^{-1}\mathbf{A}_t^\top\bar{\mathbf{X}}^\top\bar{\mathbf{X}}\Big]\mathbf{b}_t + \mathbb{E}\Big[\big(\mathbf{A}_t^\top\bar{\mathbf{X}}^\top\bar{\mathbf{X}}\mathbf{A}_t + \lambda\gamma\mathbf{I}\big)^{-1}\Big]\lambda\gamma\mathbf{x}_t \\
&\overset{(*)}{=} (\mathbf{A}_t^\top\mathbf{A}_t + \lambda\mathbf{I})^{-1}\mathbf{A}_t^\top\mathbf{b} + (\mathbf{A}_t^\top\mathbf{A}_t + \lambda\mathbf{I})^{-1}\lambda\mathbf{x}_t = \mathbf{H}^{-1}(\mathbf{x}_t)\mathbf{g}(\mathbf{x}_t),
\end{aligned}
$$

where in $(*)$ we used both formulas from Lemma 4. This concludes the proof. ∎

The proof of Theorem 7 follows analogously.

**Proof of Theorem 7** Suppose that, in Lemma 4, parameter $\gamma$ is chosen as in Definition 3 and let $\bar{\mathbf{X}} \sim \mathrm{Det}_\mu^\gamma(\mathbf{A}, \lambda)$, with $\mathbf{S} = \frac{1}{\sqrt{m}}\bar{\mathbf{X}}$. Once again, we have $\lambda' = \lambda \cdot \gamma/m$, so we can write:

$$
\mathbb{E}\big[\widehat{\mathbf{H}}_t^{-1}\big] = \mathbb{E}\Big[\big(\tfrac{m}{\gamma}\mathbf{A}_t^\top\mathbf{S}^\top\mathbf{S}\mathbf{A}_t + \lambda\mathbf{I}\big)^{-1}\Big] = \gamma\mathbb{E}\Big[\big(\mathbf{A}_t^\top\bar{\mathbf{X}}^\top\bar{\mathbf{X}}\mathbf{A}_t + \lambda\gamma\mathbf{I}\big)^{-1}\Big] \overset{(*)}{=} (\mathbf{A}_t^\top\mathbf{A}_t + \lambda\mathbf{I})^{-1},
$$

where in $(*)$ we used the second formula from Lemma 4. This concludes the proof. ∎

# B   Efficient Algorithms for Surrogate Sketches

In this section, we provide a framework for implementing surrogate sketches by relying on the algorithmic techniques from the DPP sampling literature. We then use these results to give the input-sparsity time implementation of the surrogate leverage score sampling sketch.

**Definition 4** *Given a probability measure $\mu$ over domain $\Omega$ and a kernel function $K : \Omega \times \Omega \to \mathbb{R}_{\geq 0}$, we define a determinantal point process $\mathcal{X} \sim \mathrm{DPP}_\mu(K)$ as a distribution over finite subsets of $\Omega$, such that for any $k \geq 0$ and event $E \subseteq \binom{\Omega}{k}$:*

$$
\Pr\{\mathcal{X} \in E\} \propto \frac{1}{k!}\,\mathbb{E}_{\mu^k}\Big[\mathbf{1}_{[\{x_1,\ldots,x_k\}\in E]}\det\big([K(x_i, x_j)]_{ij}\big)\Big].
$$

**Remark 12** *If $\Omega$ is the set of row vectors of an $n \times d$ matrix $\mathbf{A}$ and $K(\mathbf{a}_1, \mathbf{a}_2) = \mathbf{a}_1^\top\mathbf{a}_2$, then $\mathcal{X} \sim \mathrm{DPP}_\mu(K)$, with $\mu$ being the uniform measure over $\Omega$, reduces to a standard L-ensemble DPP [22]. In particular, let $S \sim \mathrm{DPP}(\mathbf{A}\mathbf{A}^\top)$ denote a random subset of $[n]$ sampled so that $\Pr(S) \propto \det(\mathbf{A}_S\mathbf{A}_S^\top)$. Then the set of rows of $\mathbf{A}_S$ is distributed identically to $\mathcal{X}$.*

A key property of L-ensembles, which relates them to the $\lambda$-effective dimension is that if $\mathcal{X} \sim \mathrm{DPP}_\mu(K)$ for an isotropic measure $\mu$ and $K(\mathbf{x}, \mathbf{y}) = \frac{1}{\lambda}\mathbf{x}^\top\mathbf{A}\mathbf{A}^\top\mathbf{y}$, then $\mathbb{E}[|\mathcal{X}|] = d_\lambda(\mathbf{A})$.

We next show that our determinantal design (Definition 2) can be decomposed into a DPP portion and an i.i.d. portion, which enables efficient sampling for surrogate sketches. A similar result was previously shown by [14] for their determinantal design (which is different from ours in that it is not regularized). The below result also immediately leads to the formula for the expected size of a surrogate sketch $\bar{\mathbf{X}} \sim \mathrm{Det}_\mu^\gamma(\mathbf{A}, \lambda)$ given in Section 2, which states that $\mathbb{E}[\#(\bar{\mathbf{X}})] = \gamma + d_\lambda(\mathbf{A})$.

**Lemma 13** *Let $\mu$ be a probability measure over $\mathbb{R}^n$. Given scalars $\lambda, \gamma > 0$ and a matrix $\mathbf{A} \in \mathbb{R}^{n \times d}$, let $\mathcal{X} \sim \mathrm{DPP}_\mu(K)$, where $K(\mathbf{x}, \mathbf{y}) = \frac{1}{\lambda}\mathbf{x}^\top\mathbf{A}\mathbf{A}^\top\mathbf{y}$ and $\mathbf{X} \sim \mu^M$ for $M \sim \mathrm{Poisson}(\gamma)$. Then the matrix $\bar{\mathbf{X}}$ formed by adding the elements of $\mathcal{X}$ as rows into the matrix $\mathbf{X}$ and then randomly permuting the rows of the obtained matrix, is distributed as $\mathrm{Det}_\mu^\gamma(\mathbf{A}, \lambda)$.*

**Proof** Let $E \subseteq \mathbb{R}^{t \times n}$ be an event measurable with respect to $\mu^t$. We have:

$$
\begin{aligned}
\Pr\{\bar{\mathbf{X}} \in E\} &= \sum_{s=0}^{t} \frac{\gamma^{t-s} \mathrm{e}^{-\gamma}}{(t-s)!} \cdot \frac{1}{\binom{t}{s}} \sum_{S \in \binom{[t]}{s}} \frac{\mathbb{E}_{\mu^t}[\mathbf{1}_{[\mathbf{X} \in E]} \cdot \det(\frac{1}{\lambda} \mathbf{X}_S \mathbf{A} \mathbf{A}^\top \mathbf{X}_S^\top)]}{s! \, \det(\frac{1}{\lambda} \mathbf{A}^\top \mathbf{A} + \mathbf{I})} \\
&= \frac{\gamma^t \mathrm{e}^{-\gamma}}{t!} \sum_{s=0}^{t} \sum_{S \in \binom{[t]}{s}} \frac{\mathbb{E}_{\mu^t}[\mathbf{1}_{[\mathbf{X} \in E]} \cdot \det(\frac{1}{\lambda\gamma} \mathbf{X}_S \mathbf{A} \mathbf{A}^\top \mathbf{X}_S^\top)]}{\det(\frac{1}{\lambda} \mathbf{A}^\top \mathbf{A} + \mathbf{I})} \\
&= \frac{\gamma^t \mathrm{e}^{-\gamma}}{t! \, \det(\frac{1}{\lambda} \mathbf{A}^\top \mathbf{A} + \mathbf{I})} \, \mathbb{E}_{\mu^t}\left[ \mathbf{1}_{[\mathbf{X} \in E]} \cdot \sum_{S \subseteq [t]} \det\left(\tfrac{1}{\lambda\gamma} \mathbf{X}_S \mathbf{A} \mathbf{A}^\top \mathbf{X}_S^\top\right) \right] \\
&= \frac{\gamma^t \mathrm{e}^{-\gamma}}{t! \, \det(\frac{1}{\lambda} \mathbf{A}^\top \mathbf{A} + \mathbf{I})} \mathbb{E}_{\mu^t}\left[ \mathbf{1}_{[\mathbf{X} \in E]} \cdot \det\left(\tfrac{1}{\lambda\gamma} \mathbf{A}^\top \mathbf{X}^\top \mathbf{X} \mathbf{A} + \mathbf{I}\right) \right] \\
&= \frac{\gamma^t \mathrm{e}^{-\gamma}}{t!} \cdot \frac{\mathbb{E}_{\mu^t}\left[ \mathbf{1}_{[\mathbf{X} \in E]} \cdot \det(\mathbf{A}^\top \mathbf{X}^\top \mathbf{X} \mathbf{A} + \lambda\gamma \mathbf{I}) \right]}{\det(\gamma \mathbf{A}^\top \mathbf{A} + \lambda\gamma \mathbf{I})},
\end{aligned}
$$

which concludes the proof. ∎

We now give an algorithm for sampling a surrogate of the i.i.d. row-sampling sketch where the importance sampling distribution is given by $p : [n] \to \mathbb{R}_{\geq 0}$. Here, the probability measure $\mu$ is defined so that $\mu\big(\{\frac{1}{\sqrt{p_i}} \mathbf{e}_i^\top\}\big) = p_i$ for each $i \in [n]$. The surrogate sketch $\bar{\mathcal{S}}_\mu^m(\mathbf{A}, \lambda)$ for this $\mu$ can be constructed as follows:

1. Sample set $S \subseteq [n]$ so that $\Pr\{S\} \propto \det(\frac{1}{\lambda} \mathbf{A}_S \mathbf{A}_S^\top)$, i.e., according to $\mathrm{DPP}(\frac{1}{\lambda} \mathbf{A} \mathbf{A}^\top)$.

2. Draw $M \sim \mathrm{Poisson}(\gamma)$ for $\gamma = m - d_\lambda$ and sample $i_1, ..., i_M$ i.i.d. from $p$.

3. Let $\sigma_1, ..., \sigma_{M+|S|}$ be a sequence consisting of $S$ and $i_1, ..., i_M$ randomly permuted together.

4. Then, the $i$th row of $\bar{\mathbf{S}}$ is $\frac{1}{\sqrt{m p_{\sigma_i}}} \mathbf{e}_{\sigma_i}^\top$ for $i \in [M + |S|]$.

We next present an implementation of the surrogate row sampling sketch which runs in input-sparsity time for tall matrices $\mathbf{A}$ (i.e., when $n \gg d$). The algorithm samples exactly from the surrogate sketching distribution, which is crucial for the analysis. Our algorithm is based on two recent papers on DPP sampling [10, 11], however we use a slight modification due to [7], which ensures exact sampling in input sparsity time.

---

**Algorithm 1** Sampling from $\mathrm{DPP}(\mathbf{A}\mathbf{A}^\top)$

---

1: **input:** $\mathbf{A} \in \mathbb{R}^{n \times d}$, $\mathbf{C} \in \mathbb{R}^{d \times d}$, $(\tilde{l}_1, \dots, \tilde{l}_n)$, $s = \sum_i \tilde{l}_i$, $\tilde{s} = \mathrm{tr}\big(\mathbf{C}(\mathbf{C} + \mathbf{I})^{-1}\big)$

2: **repeat**

3:    sample $u \sim \mathrm{Poisson}(r \mathrm{e}^{1/r} 2s)$

4:    sample $\rho_1, ..., \rho_u \overset{\text{i.i.d.}}{\sim} (\tilde{l}_1/s, ..., \tilde{l}_n/s)$,

5:    **for** $j = \{1, ..., u\}$ **do**

6:        compute $l_{\rho_j} = \mathbf{a}_{\rho_j}^\top (\mathbf{C} + \mathbf{I})^{-1} \mathbf{a}_{\rho_j}$

7:        sample $z_j \sim \mathrm{Bernoulli}\big(l_{\rho_j}/(2\tilde{l}_{\rho_j})\big)$

8:    **end for**

9:    set $\sigma = \{\rho_j : z_j = 1\}$, $t = |\sigma|$, $\tilde{\mathbf{A}}_{i,:} = \frac{1}{\sqrt{r l_{\sigma_i}}} \mathbf{A}_{i,:}$

10:    sample $Acc \sim \mathrm{Bernoulli}\left( \frac{\mathrm{e}^{\tilde{s}} \det(\mathbf{I} + \tilde{\mathbf{A}}_\sigma^\top \tilde{\mathbf{A}}_\sigma)}{\mathrm{e}^{t/r} \det(\mathbf{I} + \mathbf{C})} \right)$

11: **until** $Acc = \mathrm{true}$

12: **return** $\sigma_{\tilde{S}}$,    where $\tilde{S} \sim \mathrm{DPP}\big(\tilde{\mathbf{A}}_\sigma \tilde{\mathbf{A}}_\sigma^\top\big)$

---

**Lemma 14** *After an $O(\mathrm{nnz}(\mathbf{A})\log(n) + d^4\log(d))$ preprocessing step, we can construct matrix $\mathbf{C}$ and distribution $(\tilde{l}_1, ..., \tilde{l}_n)$ such that:*

$$(1 - \tfrac{1}{4\sqrt{d}})\mathbf{A}^\top\mathbf{A} \preceq \mathbf{C} \preceq (1 + \tfrac{1}{4\sqrt{d}})\mathbf{A}^\top\mathbf{A},$$

$$\frac{1}{2}\mathbf{a}_i^\top(\mathbf{C} + \mathbf{I})^{-1}\mathbf{a}_i \leq \tilde{l}_i \leq \frac{3}{2}\mathbf{a}_i^\top(\mathbf{C} + \mathbf{I})^{-1}\mathbf{a}_i \quad \text{for all } i.$$

*Moreover, with those inputs, Algorithm 1 returns $S \sim \mathrm{DPP}(\mathbf{A}\mathbf{A}^\top)$ and with probability $1 - \delta$ runs in time $O(d^4\log^2(1/\delta))$.*

**Proof** The lemma follows nearly identically as the main results of [10, 11]. One small difference is in the lines 3-9. Exploiting properties of the Poisson distribution (see proof of Theorem 2 in [7]), these lines can be rewritten as follows:

compute $\bar{s} = \sum_i l_i = \mathrm{tr}(\mathbf{A}^\top\mathbf{A}(\mathbf{C} + \mathbf{I})^{-1})$
sample $t \sim \mathrm{Poisson}(re^{1/r}\bar{s})$,
sample $\sigma_1, ..., \sigma_u \overset{\text{i.i.d.}}{\sim} (l_1/\bar{s}, ..., \tilde{l}_n/\bar{s})$

After this modification, we can follow the analysis of [11] to show the correctness of the algorithm. However, note that the above rewriting requires computing $\bar{s} = \mathrm{tr}(\mathbf{A}^\top\mathbf{A}(\mathbf{C} + \mathbf{I})^{-1})$ which takes $O(nd^2)$ time, whereas Algorithm 1 avoids that step. The rest of the time complexity analysis for both the preprocessing step and for Algorithm 1 is identical to that of [10] (see version v1 of that paper). ∎

Combining Lemmas 13 and 14, we can now prove Theorem 5, showing that a surrogate row sampling sketch can be constructed in input sparsity time.

**Proof of Theorem 5** Given matrix $\mathbf{A}$ and a row sampling distribution $p$, we can construct the surrogate row sampling sketch by using Algorithm 1 to sample from $\mathrm{DPP}(\frac{1}{\lambda}\mathbf{A}\mathbf{A}^\top)$ in time $O(\mathrm{nnz}(\mathbf{A})\log(n) + d^4\log(d))$, and then using the procedure from Lemma 13. ∎

**Computational cost of surrogate Hessian sketch** We now discuss how the surrogate Hessian sketch can be obtained using the above algorithmic framework. Recall that, letting $\lambda' = \lambda \cdot (1 - \frac{d_\lambda}{m})$, this sketch can be rewritten as follows:

$$\widehat{\mathbf{H}} = \frac{\lambda}{\lambda'}\mathbf{A}^\top\bar{\mathbf{S}}^\top\bar{\mathbf{S}}\mathbf{A} + \lambda\mathbf{I} = \frac{m}{m - d_\lambda}\mathbf{A}^\top\frac{1}{m}\bar{\mathbf{X}}^\top\bar{\mathbf{X}}\mathbf{A} + \lambda\mathbf{I} = \gamma^{-1}\mathbf{A}^\top\bar{\mathbf{X}}^\top\bar{\mathbf{X}}\mathbf{A} + \lambda\mathbf{I},$$

where $\bar{\mathbf{X}} \sim \mathrm{Det}_\mu^\gamma(\mathbf{A}, \lambda)$, with $\gamma = m - d_\lambda$ and $\mu$ being an approximate leverage score sampling distribution (which can be obtained efficiently in input sparsity time [19, 9]). Note that to obtain $\gamma$ from $m$ requires directly computing $d_\lambda$, which costs $O(nd^2)$, even more than the cost of the sketch. However, for any given choice of $\gamma \geq d$, we know that the expected sketch size $m$ satisfies $\gamma \leq m \leq \gamma + d_\lambda \leq 2\gamma$. Thus, we can get within a factor of two of any desired sketch size larger than $d$ without ever having to compute the effective dimension $d_\lambda$, which is sufficient for the complexity analysis of the Hessian sketch. Also, note that the $O(\mathrm{nnz}(\mathbf{A})\log(n))$ cost term comes only from approximating the leverage scores. In fact, many real-world data matrices have low matrix coherence [40] (a quantity that measures non-uniformity of leverage scores). Under the low coherence assumption, the uniform row sampling distribution can be used in place of approximate leverage score sampling, and as a result, surrogate Hessian sketch can be computed faster than in input sparsity time.

## C  Matrix Concentration Guarantees for Surrogate Sketches

Throughout this section, we will let $C > 0$ be a sufficiently large absolute constant. Also, we use the notation $\mathbf{H} = \mathbf{A}^\top\mathbf{A} + \lambda\mathbf{I}$ for the Hessian, and we let $\kappa$ be the condition number of $\mathbf{H}$. Consider the determinantal design $\mathrm{Det}_\mu^\gamma(\mathbf{A}, \lambda)$ where the probability measure $\mu$ is defined so that $\mu(\{\frac{1}{\sqrt{p_i}}\mathbf{e}_i^\top\}) = p_i$ and $p_i \geq \frac{1}{2}\mathbf{a}_i^\top\mathbf{H}^{-1}\mathbf{a}_i/d_\lambda$, for each $i \in [n]$.

We analyze the concentration properties of the random matrix of the form:

$$\mathbf{Z} = \mathbf{H}^{-\frac{1}{2}}(\gamma^{-1}\mathbf{A}^\top\bar{\mathbf{X}}^\top\bar{\mathbf{X}}\mathbf{A} + \lambda\mathbf{I})\mathbf{H}^{-\frac{1}{2}} \qquad \text{for} \quad \bar{\mathbf{X}} \sim \mathrm{Det}_\mu^\gamma(\mathbf{A}, \lambda).$$

While standard matrix concentration results apply to sums of independent matrices, recent work of [23] extended these results to a class of non-i.i.d. distributions known as Strongly Rayleigh measures. Since determinantal point processes are Strongly Rayleigh, our determinantal designs can also be expressed this way, which allows us to obtain the following guarantee.

**Lemma 15** *If $\gamma \geq Cd_\lambda \eta^{-2} \log(n/\delta)$ for $\eta \in (0,1)$, then with probability $1-\delta$ we have $\|\mathbf{Z}-\mathbf{I}\| \leq \eta$.*

We first quote the two matrix concentration results which we use to prove Lemma 15. The more standard result applies to sums of independent random matrices, and can be stated as follows.

**Theorem 16 ([38])** *Consider a finite sequence $\mathbf{Y}_1, \mathbf{Y}_2, \ldots$ of independent random positive semi-definite $d \times d$ matrices that satisfy $\|\mathbf{Y}_i\| \leq R$ and $\mu_{\min}\mathbf{I} \preceq \mathbb{E}[\sum_i \mathbf{Y}_i] \preceq \mu_{\max}\mathbf{I}$. Then,*

$$\Pr\left(\lambda_{\max}\left(\sum_i \mathbf{Y}_i\right) \geq (1+\epsilon)\mu_{\max}\right) \leq d\exp\left(-\frac{\epsilon^2 \mu_{\max}}{3R(1+\epsilon)}\right) \quad \text{for } \epsilon > 0,$$

$$\Pr\left(\lambda_{\min}\left(\sum_i \mathbf{Y}_i\right) \leq (1-\epsilon)\mu_{\min}\right) \leq d\exp\left(-\frac{\epsilon^2 \mu_{\min}}{2R}\right) \quad \text{for } \epsilon \in (0,1).$$

The other matrix concentration result applies to sums of non-independent random matrices, and it can be viewed as a partial extension of the above theorem, except with an additional logarithmic factor.

**Theorem 17 ([23])** *Suppose $(\xi_1, \ldots, \xi_n) \in \{0,1\}^n$ is a random vector whose distribution is $k$-homogeneous (i.e., exactly $k$-sparse almost surely) and Strongly Rayleigh. Given $d \times d$ p.s.d. matrices $\mathbf{C}_1, \ldots, \mathbf{C}_n$ such that $\|\mathbf{C}_i\| \leq R$ and $\|\mathbb{E}[\sum_i \xi_i \mathbf{C}_i]\| \leq \mu$, for any $\epsilon > 0$ we have:*

$$\Pr\left(\left\|\sum_i \xi_i \mathbf{C}_i - \mathbb{E}\left[\sum_i \xi_i \mathbf{C}_i\right]\right\| \geq \epsilon\mu\right) \leq d\exp\left(-\frac{\epsilon^2 \mu}{R(\log k + \epsilon)} \cdot \Theta(1)\right).$$

**Proof of Lemma 15** To perform the analysis, we separate the matrix $\bar{\mathbf{X}}$ into two parts as described in Lemma 13: the rows coming from $\mathrm{DPP}(\frac{1}{\lambda}\mathbf{A}\mathbf{A}^\top)$, and the remaining i.i.d. rows. Denote the former as $\bar{\mathbf{X}}_{\mathrm{DPP}}$ and the latter as $\bar{\mathbf{X}}_{\mathrm{IID}}$. The i.i.d. part can be analyzed via the matrix concentration result for independent matrices (Theorem 16) by setting

$$\mathbf{Y}_i = \mathbf{H}^{-\frac{1}{2}}\left(\frac{1}{\gamma p_{j_i}}\mathbf{a}_{j_i}\mathbf{a}_{j_i}^\top + \frac{\lambda}{\gamma}\mathbf{I}\right)\mathbf{H}^{-\frac{1}{2}},$$

where $M$ is the number of rows in $\bar{\mathbf{X}}_{\mathrm{IID}}$, $\mathbf{a}_j^\top$ denotes the $j$th row of $\mathbf{A}$ and $j_i$ is the row index sampled according to the approximate $\lambda$-ridge leverage score $\{p_j\}$ distribution. Since the number of rows in $\bar{\mathbf{X}}_{\mathrm{IID}}$ is a random variable $M \sim \mathrm{Poisson}(\gamma)$, to apply Theorem 16 we first condition on $M$. Note that we have $\sum_{i=1}^M \mathbf{Y}_i = \mathbf{H}^{-\frac{1}{2}}(\frac{1}{\gamma}\mathbf{A}^\top \bar{\mathbf{X}}_{\mathrm{IID}}^\top \bar{\mathbf{X}}_{\mathrm{IID}}\mathbf{A} + \frac{M}{\gamma}\lambda\mathbf{I})\mathbf{H}^{-\frac{1}{2}}$ and $\mathbb{E}[\sum_{i=1}^m \mathbf{Y}_i \mid M] = \frac{M}{\gamma}\mathbf{I}$. Furthermore, using the assumption on the probabilities $p_j$, we have $\|\mathbf{Y}_i\| \leq \frac{3}{\gamma}$. Thus, Theorem 16 implies that (conditioned on $M$):

$$\Pr\left(\left\|\mathbf{H}^{-\frac{1}{2}}(\gamma^{-1}\mathbf{A}^\top \bar{\mathbf{X}}_{\mathrm{IID}}^\top \bar{\mathbf{X}}_{\mathrm{IID}}\mathbf{A} + \frac{M}{\gamma}\lambda\mathbf{I})\mathbf{H}^{-\frac{1}{2}} - \frac{M}{\gamma}\mathbf{I}\right\| \geq \eta \cdot \frac{M}{\gamma}\right) \leq 2d\exp\left(-\eta^2 M/12\right). \quad (4)$$

For sufficiently large constant $C$, standard tail bounds for the Poisson distribution (e.g., Theorem 1 in [8]) imply that with probability $1 - \delta$, we have $|M - \gamma| \leq \eta\gamma$ and, conditioned on this event, (4) can be bounded by $\delta$. Putting this together, we conclude that with probability $1 - 2\delta$:

$$\|\mathbf{H}^{-\frac{1}{2}}(\tfrac{1}{\gamma}\mathbf{A}^\top \bar{\mathbf{X}}_{\mathrm{IID}}^\top \bar{\mathbf{X}}_{\mathrm{IID}}\mathbf{A} + \lambda\mathbf{I})\mathbf{H}^{-\frac{1}{2}} - \mathbf{I}\|$$
$$= \|\mathbf{H}^{-\frac{1}{2}}(\tfrac{1}{\gamma}\mathbf{A}^\top \bar{\mathbf{X}}_{\mathrm{IID}}^\top \bar{\mathbf{X}}_{\mathrm{IID}}\mathbf{A} + \tfrac{M}{\gamma}\lambda\mathbf{I})\mathbf{H}^{-\frac{1}{2}} - \tfrac{M}{\gamma}\mathbf{I} + (1 - \tfrac{M}{\gamma})\lambda\mathbf{H}^{-1} + (\tfrac{M}{\gamma} - 1)\mathbf{I}\|$$
$$\leq \|\mathbf{H}^{-\frac{1}{2}}(\tfrac{1}{\gamma}\mathbf{A}^\top \bar{\mathbf{X}}_{\mathrm{IID}}^\top \bar{\mathbf{X}}_{\mathrm{IID}}\mathbf{A} + \tfrac{M}{\gamma}\lambda\mathbf{I})\mathbf{H}^{-\frac{1}{2}} - \tfrac{M}{\gamma}\mathbf{I}\| + 2\left|\tfrac{M}{\gamma} - 1\right| \leq 4\eta,$$

where we used that $\|\lambda\mathbf{H}^{-1}\| \leq 1$ and $\eta\frac{M}{\gamma} \leq \eta(1+\eta) \leq 2\eta$. From this it immediately follows that:

$$\mathbf{Z} \succeq \mathbf{H}^{-\frac{1}{2}}(\gamma^{-1}\mathbf{A}^\top \bar{\mathbf{X}}_{\mathrm{IID}}^\top \bar{\mathbf{X}}_{\mathrm{IID}}\mathbf{A} + \lambda\mathbf{I})\mathbf{H}^{-\frac{1}{2}} \succeq (1 - 4\eta)\mathbf{I}. \quad (5)$$

Next, to bound $\mathbf{Z}$ from above we must analyze the contribution of $\bar{\mathbf{X}}_{\mathrm{DPP}}$. Note that the matrix concentration result of [23] applies to homogeneous Strongly Rayleigh distributions (i.e., where the sample size is fixed) whereas $S \sim \mathrm{DPP}(\frac{1}{\lambda}\mathbf{A}\mathbf{A}^\top)$ is non-homogeneous, because the size of $S$ is randomized. However, we can apply a standard transformation known as symmetric homogenization to transform $S$ from a non-homogeneous distribution over subsets of $[n]$ into $\tilde{S}$ which is an $n$-homogeneous distribution over subsets of $[2n]$, in such a way that $\tilde{S} \cap [n]$ is distributed identically to $S$ and is Strongly Rayleigh (see Definition 2.12 in [6]). Now, we can apply Theorem 17 with $\mathbf{C}_i = \frac{1}{p_i}\mathbf{H}^{-\frac{1}{2}}\mathbf{a}_i\mathbf{a}_i^\top\mathbf{H}^{-\frac{1}{2}}$ for $i \in [n]$ and $\mathbf{C}_i = \mathbf{0}$ for $i > n$, and defining $\xi_i = \mathbf{1}_{[i \in \tilde{S}]}$. Note that $\sum_i \xi_i \mathbf{C}_i = \mathbf{H}^{-\frac{1}{2}}\mathbf{A}^\top\bar{\mathbf{X}}_{\mathrm{DPP}}^\top\bar{\mathbf{X}}_{\mathrm{DPP}}\mathbf{A}\mathbf{H}^{-\frac{1}{2}}$ and $\|\mathbf{C}_i\| \leq 2$. To obtain the expectation of the sum, we use the fact that the marginal probabilities of a DPP are the ridge leverage scores, i.e., $\Pr(i \in S) = \mathbf{a}_i^\top(\mathbf{A}^\top\mathbf{A} + \lambda\mathbf{I})^{-1} =: l_i(\lambda)$. We obtain that:

$$\mathbb{E}\Big[\sum_i \xi_i \mathbf{C}_i\Big] = \sum_i \frac{l_i(\lambda)}{p_i}\mathbf{H}^{-\frac{1}{2}}\mathbf{a}_i\mathbf{a}_i^\top\mathbf{H}^{-\frac{1}{2}} \preceq 2d_\lambda\mathbf{H}^{-\frac{1}{2}}\mathbf{A}^\top\mathbf{A}\mathbf{H}^{-\frac{1}{2}} \preceq 2d_\lambda\mathbf{I}.$$

Setting $\epsilon = C\log(n/\delta)/2$ for large enough $C$, Theorem 17 implies that with probability $1 - \delta$:

$$\|\mathbf{H}^{-\frac{1}{2}}\mathbf{A}^\top\bar{\mathbf{X}}_{\mathrm{DPP}}^\top\bar{\mathbf{X}}_{\mathrm{DPP}}\mathbf{A}\mathbf{H}^{-\frac{1}{2}}\| \leq Cd_\lambda\log(n/\delta).$$

Note that $\gamma^{-1}Cd_\lambda\log(n/\delta) \leq \eta$, so with probability $1 - 3\delta$:

$$\mathbf{Z} = \frac{1}{\gamma}\mathbf{H}^{-\frac{1}{2}}\mathbf{A}^\top\bar{\mathbf{X}}_{\mathrm{DPP}}^\top\bar{\mathbf{X}}_{\mathrm{DPP}}\mathbf{A}\mathbf{H}^{-\frac{1}{2}} + \mathbf{H}^{-\frac{1}{2}}(\frac{1}{\gamma}\mathbf{A}^\top\bar{\mathbf{X}}_{\mathrm{IID}}^\top\bar{\mathbf{X}}_{\mathrm{IID}}\mathbf{A} + \lambda\mathbf{I})\mathbf{H}^{-\frac{1}{2}} \preceq (1 + 5\eta)\mathbf{I}. \quad (6)$$

Combining (5) and (6) we get $\|\mathbf{Z} - \mathbf{I}\| \leq 5\eta$. Adjusting the constants concludes the proof. ∎

Next, we we use the above matrix concentration result to derive moment bounds for the spectral norm of the random matrix $\mathbf{Z}^{-1} - \mathbb{E}[\mathbf{Z}^{-1}]$. Note that by Lemma 4 we have $\mathbb{E}[\mathbf{Z}^{-1}] = \mathbf{I}$. Also, recall that we use $\kappa$ to denote the condition number of $\mathbf{H}$.

**Lemma 18** *If $\gamma \geq Cd_\lambda\eta^{-2}p\log(n\kappa/\eta)$ then:*

$$\mathbb{E}\Big[\|\mathbf{Z}^{-1} - \mathbb{E}[\mathbf{Z}^{-1}]\|^p\Big]^{\frac{1}{p}} \leq \eta.$$

**Proof** From Lemma 15 we know that if $\gamma \geq Cd_\lambda\eta^{-2}\log(n/\delta)$, then with probability $1 - \delta$ we have $\|\mathbf{Z} - \mathbf{I}\| \leq \eta$, which also implies that $\|\mathbf{Z}^{-1} - \mathbb{E}[\mathbf{Z}^{-1}]\| \leq \eta$, since $\mathbb{E}[\mathbf{Z}^{-1}] = \mathbf{I}$. Also, note that $\|\mathbf{Z}^{-1} - \mathbb{E}[\mathbf{Z}^{-1}]\| \leq \kappa$ almost surely. It follows that:

$$\mathbb{E}\Big[\|\mathbf{Z}^{-1} - \mathbb{E}[\mathbf{Z}^{-1}]\|^p\Big] \leq \eta^p + \delta\kappa^p.$$

Letting $\delta = \eta/\kappa^p$, we obtain that $\mathbb{E}[\|\mathbf{Z}^{-1} - \mathbb{E}[\mathbf{Z}^{-1}]\|^p]^{\frac{1}{p}} \leq 2\eta$. Adjusting the constants appropriately, we obtain the desired result. ∎

Using this moment bound, we can show that the average of the inverses of $q$ i.i.d. copies of $\mathbf{Z}$, denoted $\mathbf{Z}_1, ..., \mathbf{Z}_q$ exhibit concentration around the mean with high probability. We first quote a Khintchine/Rosenthal type inequality from [21], bounding the moments of a sum of random matrices.

**Lemma 19 ([21])** *Suppose that $p \geq 2$ and $r = \max\{p, 2\log d\}$. Consider a finite sequence $\{\mathbf{Y}_k\}$ of independent, symmetrically random, self-adjoint matrices with dimension $d \times d$. Then,*

$$\mathbb{E}\Big[\|\sum_k \mathbf{Y}_k\|^p\Big]^{\frac{1}{p}} \leq \sqrt{er}\,\Big\|\sum_k \mathbb{E}[\mathbf{Y}_k^2]\Big\|^{\frac{1}{2}} + 2er\,\mathbb{E}\big[\max_k \|\mathbf{Y}_k\|^p\big]^{\frac{1}{p}}.$$

We are now ready to prove Lemma 8. We state here a more precise version of the lemma. The proof follows by combining Lemmas 18 and 19, following a standard conversion from moment bounds to high-probability guarantees (our proof is based on the proof of Corollary 13 from [15]).

**Lemma 20 (restated Lemma 8)** *If $\gamma \geq Cd_\lambda\eta^{-2}\log^4\big(\frac{n\kappa}{\eta\delta}\big)$ then with probability $1 - \delta$, we have:*

$$\Big\|\frac{1}{q}\sum_{k=1}^q (\mathbf{Z}_k^{-1} - \mathbb{E}[\mathbf{Z}_k^{-1}])\Big\| \leq \frac{\eta}{\sqrt{q}}.$$

**Proof** From Lemma 18, we know that if $\gamma \geq Cd_\lambda\tilde{\eta}^{-2}p\log(n\kappa/\tilde{\eta})$, then for any $2 \leq s \leq p$ we have $\mathbb{E}[\|\mathbf{Z}_k^{-1} - \mathbb{E}[\mathbf{Z}_k^{-1}]\|^s]^{\frac{1}{s}} \leq \tilde{\eta}$. Following a standard symmetrization argument, we can write the following:

$$\mathbb{E}\Big[\big\|\frac{1}{q}\sum_{k=1}^{q}(\mathbf{Z}_k^{-1} - \mathbb{E}[\mathbf{Z}_k^{-1}])\big\|^p\Big]^{\frac{1}{p}} \leq 2 \cdot \mathbb{E}\Big[\big\|\sum_{k=1}^{q}\frac{r_k}{q}(\mathbf{Z}_k^{-1} - \mathbb{E}[\mathbf{Z}_k^{-1}])\big\|^p\Big]^{\frac{1}{p}},$$

where $r_k$ are independent Rademacher random variables. We now apply Lemma 19 to the symmetrically random matrices $\mathbf{Y}_k = \frac{r_k}{q}(\mathbf{Z}_k^{-1} - \mathbb{E}[\mathbf{Z}_k^{-1}])$, obtaining:

$$\mathbb{E}\Big[\big\|\sum_{k=1}^{q}\mathbf{Y}_k\big\|^p\Big]^{\frac{1}{p}} \leq \sqrt{er}\,\big\|\sum_k\mathbb{E}[\mathbf{Y}_k^2]\big\|^{\frac{1}{2}} + 2er\,\mathbb{E}\big[\max_k\|\mathbf{Y}_k\|^p\big]^{\frac{1}{p}}$$

$$\leq \sqrt{\frac{er}{q}}\cdot\mathbb{E}\big[\|\mathbf{Z}_k^{-1} - \mathbb{E}[\mathbf{Z}_k^{-1}]\|^2\big]^{\frac{1}{2}} + \frac{2er}{\sqrt{q}}\mathbb{E}\big[\|\mathbf{Z}_k^{-1} - \mathbb{E}[\mathbf{Z}_k^{-1}]\|^p\big]^{\frac{1}{p}} \leq \frac{C'p}{\sqrt{q}}\cdot\tilde{\eta},$$

for $p \geq 2\log(d)$. Finally, following the proof of Corollary 13 of [15], we use Markov's inequality with $\alpha = \frac{\eta}{\sqrt{q}}$, $\tilde{\eta} = \frac{\eta}{4C'p}$ and $p = 2\lceil\max\{\log(1/\delta), \log(d)\}\rceil$:

$$\Pr\left(\big\|\frac{1}{q}\sum_{k=1}^{q}(\mathbf{Z}_k^{-1} - \mathbb{E}[\mathbf{Z}_k^{-1}])\big\| \geq \alpha\right) \leq \alpha^{-p}\cdot\mathbb{E}\Big[\big\|\frac{1}{q}\sum_{k=1}^{q}(\mathbf{Z}_k^{-1} - \mathbb{E}[\mathbf{Z}_k^{-1}])\big\|^p\Big] \leq \left(\frac{2C'p\tilde{\eta}}{\alpha\sqrt{q}}\right)^p \leq \delta,$$

which completes the proof. ∎

Finally, note that the sketched Hessian matrix $\widehat{\mathbf{H}}_t$, defined in Section 3.2 and used in Lemma 8, where recall that $\lambda' = \lambda\cdot(1 - \frac{d_\lambda}{m})$, satisfies the following:

$$\widehat{\mathbf{H}}_t = \frac{m}{m - d_\lambda}\mathbf{A}^\top\frac{1}{m}\bar{\mathbf{X}}^\top\bar{\mathbf{X}}\mathbf{A} + \lambda\mathbf{I} = \gamma^{-1}\mathbf{A}^\top\bar{\mathbf{X}}^\top\bar{\mathbf{X}}\mathbf{A} + \lambda\mathbf{I} = \mathbf{H}^{\frac{1}{2}}\mathbf{Z}\mathbf{H}^{\frac{1}{2}},$$

so the approximation guarantee from Lemma 8 follows immediately from Lemma 20 by setting $\eta = \frac{1}{\sqrt{\alpha}}$ and assuming that $m \leq n$.

## D  Additional Numerical Results

In this section, we present additional numerical results. Figure 6 complements Figures 1 and 2, demonstrating on two additional datasets that rescaling the local regularizer helps reduce the estimation error of distributed averaging for Gaussian sketch, Rademacher sketch, uniform sampling, and the surrogate sketch. Figure 7 shows the comparison of surrogate sketch, uniform sampling with unweighted averaging, and determinantal averaging, on the Boston housing prices dataset, similarly to Figure 4 but for different sketch sizes. We observe that the surrogate sketch still empirically outperforms the other methods in the case of different sketch sizes.

We next give the exact form of the optimization problem solved in the experiments of Figure 3 in the main body of the paper. The problem solved in Figure 3 is a logistic regression problem with $\ell_2$ regularization:

$$\text{minimize}_{\mathbf{x}}\ -\sum_{i=1}^{n}\big(b_i\log(p_i) + (1 - b_i)\log(1 - p_i)\big) + \frac{\lambda}{2}\|\mathbf{x}\|^2, \tag{7}$$

where $\mathbf{p} \in \mathbb{R}^n$ is defined such that $p_i = 1/(1 + \exp(-\mathbf{a}_i^\top\mathbf{x}))$. Here $\mathbf{a}_i^\top$ represents the $i$'th row of the data matrix $\mathbf{A} \in \mathbb{R}^{n\times d}$. The output vector is denoted by $\mathbf{b} \in \mathbb{R}^n$.

In addition to the plots in Figure 3, we include more experimental results for the distributed Newton sketch algorithm. Instead of regularized logistic regression, we consider a different convex optimization problem which has inequality constraints $\|\mathbf{A}\mathbf{x}\|_\infty \leq t$ and a quadratic objective $\|\mathbf{x} - \mathbf{c}\|^2$. Problems of this form can be converted into an unconstrained optimization problem using the log-barrier method [3] to obtain:

$$\text{minimize}_{\mathbf{x}}\ -\sum_{i=1}^{n}\log(-\mathbf{a}_i^\top\mathbf{x} + t) - \sum_{i=1}^{n}\log(\mathbf{a}_i^\top\mathbf{x} + t) + \lambda\|\mathbf{x}\|^2 - 2\lambda\mathbf{c}^\top\mathbf{x} + \lambda\|\mathbf{c}\|^2, \tag{8}$$

(a) diabetes

(b) monks-1

Figure 6: Estimation error against the number of averaged outputs for two different datasets; diabetes and monks-1. The dotted lines show the error when the local regularizer $\lambda'$ is picked using the expression in Theorem 1 whereas the straight lines correspond to using the value of the global regularizer as the local regularizer. These plots show the performances of the same sketches as Figure 1 on two additional datasets. Problem parameters are as follows: Plot a: $n = 440$, $d = 10$, $m = 20$, $\lambda = 1$. Plot b: $n = 124$, $d = 6$, $m = 20$, $\lambda = 100$.

(a) $m = 20$

(b) $m = 100$

Figure 7: Estimation error of the *surrogate sketch*, against uniform sampling with *unweighted averaging* [40] and *determinantal averaging* [15]. The plots in this figure follow the same setting as Figure 4 except for the sketch size. In these plots, different sketch sizes are used, which are given in the captions of each plot.

where $t > 0$, $\mathbf{c} \in \mathbb{R}^d$, and $\mathbf{a}_i$ is the $i$'th row of the data matrix $\mathbf{A} \in \mathbb{R}^{n \times d}$. Figure 8 shows the normalized error against iterations for the distributed Newton sketch algorithm when it is used to solve the problem (8). We note that rescaling the local regularizer as in Theorem 1 leads to speedups in convergence for the problem (8) for Gaussian sketch and surrogate sketch.

We note that the step sizes for the distributed Newton sketch algorithm in the experiments of both Figure 3 and Figure 8 have been determined using backtracking line search. The same set of backtracking line search parameters has been used in all of the distributed Newton sketch experiments: Initial step size is set to $\alpha_0 = 1$, update parameter is $\tau = 2$ (where the step size updates are of the form $\alpha \leftarrow \alpha/\tau$), and the control parameter is $c = 0.1$.

Figure 9 shows for various sketches that choosing the local regularizer as stated in Theorem 1 in fact leads to the lowest estimation error possible that one could hope to achieve by optimizing over the local regularization parameter. The dotted blue lines show the estimation error if we were to choose the local regularizer to be values from 1 to 10. The straight red line shows the error when the local regularizer $\lambda'$ is picked using the expression in Theorem 1. We observe that in the regime where the number of workers is high, the lowest estimation error is achieved by the red line. We also see that this empirical observation is true for not only the surrogate sketch but also Gaussian sketch and uniform sampling.

(a) $q = 20$ workers

(b) $q = 100$ workers

Figure 8: Distributed Newton sketch algorithm. The problem being solved is an inequality constrained optimization problem, which is transformed into an unconstrained problem using the log-barrier method, as given in (8). Data dimensions: $n = 500$, $d = 50$, $m = 100$. $\lambda = 10$. The step size for distributed Newton sketch updates has been determined via backtracking line search with parameters $\tau = 2$, $c = 0.1$, $a_0 = 1$. The dotted lines show the error when the local regularizer $\lambda'$ is picked using the expression in Theorem 1 whereas the straight lines correspond to using the value of the global regularizer as the local regularizer.

(a) Gaussian

(b) Uniform

(c) Surrogate sketch

Figure 9: Estimation error against the number of workers for various sketches for the Boston housing prices dataset. The red line shows the error for the debiased version according to Theorem 1 and the dotted blue lines correspond to using different local regularization parameter values from $[1, 10]$. All of the curves are averaged over 25 independent trials. Sketch size is $m = 20$ and the global regularization parameter is $\lambda = 10$.