[Reviews · NeurIPS 2020]

Review 1

Summary and Contributions: This paper proposes a sketching method based on the determinantal point process for speeding up distributed second order optimization methods. The sketching method allows for closed form computation of the bias, which can be removed by adjusting the regularization parameter. This adjustment of the regularization parameter also has empirical benefits for other (simpler) sketching methods. -------- update after author response --------- I thank the authors for addressing my comments. I particularly look forward to seeing how you can avoid explicitly/precisely computing d_lambda.

Strengths: This paper makes a nice contribution in terms of designing a new sketching method that allows for faster convergence than prior work in theory (I am less convinced about the practical implications, see below). The adjustment of the regularization parameter is also interesting and empirically appears to have practical benefits even for other sampling methods.

Weaknesses: I am somewhat concerned about the implementability of this method. Specifically, as far as I can tell, knowledge of the effective dimension d_lambda seems to be necessary to do the regularization parameter adjustment, and also to implement the surrogate sketch. I am not sure, but I am concerned that computing this d_lambda would require an SVD, in which case you might as well just do the exact Newton step. Is there a more clever way of computing d_lambda? Or does the method still work with only a rough estimate of d_lambda? I think these considerations should be addressed. Also, from a practical perspective, looking at Table 1 I'm wondering if there is such a huge difference between the uniform sampling approach of [25] and your method? In particular, a logarithmic number of iterations of [25]'s method with alpha = O(1) would suffice to reach high accuracy. Of course, your method may require only a single iteration, but the size of the sketch is much larger and computing the surrogate sketch appears to be quite a bit more complicated and requires Omega(d^4) time. It seems that the main practical benefit of your method therefore only comes in a large q and very high accuracy regime? Or am I missing something here?

Correctness: I believe so.

Clarity: The paper is very well written, and presents the ideas in a clear and easy to understand format.

Relation to Prior Work: The authors do a good job of situating this work in the context of previous work.

Reproducibility: Yes

Additional Feedback:


Review 2

Summary and Contributions: This paper considers debiasing subsampled ridge regression by 1) subsampling via a combination of leverage score and DPP sampling (using a `surrogate sketch' framework, which debiasing any sketching distribution using DPP sampling) 2) making a simple adjustment of the ridge parameter in the regression problem. Together these steps ensure that the expectation of the sketched solution is exactly the optimal solution to the problem. The sampling step can be performed reasonably quickly (in input sparsity time, but slower than say just leverage score sampling as the runtime involves a d^4 factor when the design matrix A is n x d). The technique is motivated by an application to distributed optimization: the subsampling plus ridge adjustment technique can be used to compute an unbiased estimate of an exact Newton step for convex optimization using a sketch. If many servers compute sketched Newton steps which are unbiased, averaging them together gives a better approximation of the true Newton step and so faster convergence. This is not the case for existing sketched Newton's algorithms, as no matter how many sketched copies are averaged, a bias term still bounds the different between the distributedly computed step and the true update.

Strengths: This seems to be an interesting use of DPPs and the fact that, as compared to independent sampling techniques like leverage scores, you can exactly understand the expectation of the inverse of the sampled A^T A matrix. In general even the simple example of debiasing ridge regression is nice. I liked how, while the formula for adjusting the ridge score only holds in theory for the leverage score + DPP sketch proposed, it was evaluated and seemed to give better performance for other standard sketching techniques. The idea of a surrogate sketch is nice and give a framework for debiasing any sketching distribution. I wondered if more general properties of this sketch could be stated. E.g., if the sketching distribution gives a spectral approximation to A^T A then does a result analogous to Lemma 8 ( that the sketched inverse is close to the true inverse) always hold? Or does this just hold when the initial sketch is via leverage score sampling.

Weaknesses: I had a few concerns/confusions about the algorithmic motivations. 1. To debias the sketch, it seems that one needs to know the statistical dimension d_lambda of the design matrix A. This can't be computed accurately without basically the same runtime as required to solve the ridge regression problem in the first place. Thus is seems there will be some bias, possibly defeating the purpose of the approach. I couldn't find this issue discussed in the paper. A similar issue arrises when actually computing the surrogate sketch. 2. The cost to hat H in Theorem 2 seems slower than known runtimes for just fully solving the system? E.g. via sparse sketching, Clarkson and Woodruff 2013, you can compute a constant factor preconditioner in nnz(A)+d^3 time and then solve the system to epsilon accuracy in log(1/epsilon) iterations each costing O(nnz(A)+d^2) time. So why doesn't a single server just directly solve the system and make na exact newton step instead of computing the Hessian sketch?

Correctness: I did not carefully check the proofs however generally things look reasonable and the definitions are careful, theorem are clear, etc. Very clear and well written paper. Easy to follow.

Clarity: Very clear and well written paper. Easy to follow.

Relation to Prior Work: lot of the technical work seems close to that of [9], with the mentioned difference being that in this work there is regularization. This doesn't seem to be a huge difference to me. Maybe the authors can mention the main challenges in dealing with regularization? Or do the proofs of [9] pretty much go through?

Reproducibility: Yes

Additional Feedback: - I was confused about the comparison to [10]. It says the tradeoff is that you require alpha > d. But in Table 1 it also says that [10] requires alpha > d. If there is a really a d factor loss, why not just run [10] with alpha' = alpha'*d and get the same convergence rate as given in the current paper? - What is capital E in Def 2? Just any set of outcomes? Why is is stated like this? - For Def 3, how can you set m if you don't know the stat dimension? If you just have an estimate or an upper bound say of d, what do you do? This same issue is coming up in the steps starting at line 441 of the appendix. Does Algo 1 perform these steps without having to actually compute d_lambda? It seems that Algo 1 just performs step 1 and not 2-4. Is that correct? - Is a surrogate sketch for a general random projection useful? I like that the results are stated generally, but was wondering if there was a use for this. Rebuttal: Thanks to the authors for clarifying a number of points. My biggest concern is still that the runtime in Theorem 2 is nnz(A)*log n + tilde O(alpha*d^3). There are already centralized algorithms that solve the regression problem to accuracy epsilon in nnz(A)*log(1/epsilon) + tilde O(d^3) time. The authors point out that the given algorithm can be faster. But it seems that this is only if one is trying to get super small error, so log(1/epsilon) > log n but the number of machines q is large enough so log(kappa/epsilon)/log(alpha*q) is much less than log 1/epsilon. This doesn't seem to be a very compelling scenario. As it means that we would need to have a*q >> 1/epsilon and so a* q > n. So basically you need a huge number of machines, or if you don't must pay polynomial factors in alpha = n/# machines. Even so, the paper was well written, the surrogate sketch idea is nice. So I remain on the fence with this one.


Review 3

Summary and Contributions: This work studies sketching the Newton’s method in the big n regime (the number of data is much larger than the number of parameter). To this end, the authors propose a sketch based on determinantal point process, and equip it with a refined regularization hyperparameter to eliminate the bias caused by inverting the sketched Hessian. Convergence guarantee is also established for the sketched Newton’s method, which, however, depends on the condition number.

Strengths: This work gives an unbiased sketch for newton’s method with theoretical analysis.

Weaknesses: The convergence rate of the sketch Newton’s method depends on the condition number.

Correctness: yes

Clarity: yes

Relation to Prior Work: Yes. I would recommend to discuss recent advance for stochastic Newton’s method under big n regime, for example [1].

Reproducibility: Yes

Additional Feedback: While sketching a first order method is relatively easy, sketching a second order method, e..g, Newton’s method is quite challenging. This is due to the inverse of a sketched Hessian can be biased. To address this problem, this work adopts a determinant preserving sketch and surrogates the regularization parameter to establish an unbiased sketching for Newton’s method. I think this is a very interesting way to improve Newtons’ method under big n regime, where the number of data is much bigger than the number of parameters. Concerns: - Compared to the first order methods, one of the biggest benefit of the Newton’s method is that the convergence does not depend on the condition number. However, after sketching the Newton’s method, the authors could only provide a convergence rate depending on the condition number (in a square root order) as Theorem 10. I thus quite doubt the practical significance of the sketched Newton’s method. I would recommend to also compare numerical performance with first order methods for example GD and sketched GD. - To my understanding, your construction heavily relies on the linear structure of the model (Ax). Can you comment on extending your techniques to the non-linear models?


Review 4

Summary and Contributions: This paper introduced surrogate sketching and scaled regularization to address the bias issue in distributed second order optimization. The ideas are backed by both theoretical analysis as well as empirical evaluations. ======= Post-rebuttal edit: Thanks for the response. Please incorporate the discussions on the assumption alpha>d if accepted.

Strengths: This paper addresses the issue of bias correction in distributed second order optimization, where prior method [25] suffers from a bias issue, so the performance gain starts to diminish after the number of works q is above certain threshold. Compared with the result in [10], this work improves the rate by a factor of d according to Table 2. This result is of interest to the NeuriPS community since distributed optimization is an important topic. I particularly like the idea of scaled regularization which is quite simple and elegant, and can be easily adopted into practice.

Weaknesses: The theoretical results seem have some limitations: - the assumption alpha>d is quite restrictive. In the high-dimensional setting where m/d= alpha, it is typically assumed that both m and d go to infinity while keeping alpha fixed as a constant. This work does not follow this regime and falls back to the low-dimensional regime m~alpha d>d^2, which limits the appeal of the theoretical result. - again under the assumption alpha>d, the rate of [25] becomes (1/alpha q+ 1/d^2) which means the bias only becomes significant when q is greater than d. Given d is typically large, it is unclear why this becomes a concern in this regime. - the result (theorem 10) in the general case seems to be much worse than the least-squares case, at least by a factor of sqrt{kappa}. There is no discussion about why the results in the general case degenerates compared to the least-squares case, and how it compares with prior result. The authors should also evaluate the performance as a function of the condition number.

Correctness: The results and methods are correct to the extent being checked.

Clarity: The paper is very well-written and reads clearly. Overall it did a good job in covering various concepts necessary in this topic in a fairly short page limit.

Relation to Prior Work: Table 1 only compared with prior results for the quadratic case (regularized least squares). It is important also to include a discussion on the general convex case.

Reproducibility: Yes

Additional Feedback: Overall the paper made some contributions to distributed optimization by introducing surrogate sketch and scaled regularization. The drawbacks are the assumption alpha>d and the relative weak result in the general convex case. It is unclear if the authors can alleviate such shortcomings, but at least I expect some discussions in the final version if it is accepted.

[Author Response · NeurIPS 2020]

We thank all the reviewers for their feedback and suggestions. We first address some commonly raised points:

**1. Computing the effective dimension** $d_\lambda$ (raised by Reviewers 1 and 2). In the case of the surrogate leverage score sampling sketch, the effective dimension does not need to be explicitly calculated. In particular, this means that the input sparsity time algorithm from Theorem 2 does *not* require any knowledge of $d_\lambda$ (this applies to both sketching and scaling). When using scaled regularization with a standard sketch (e.g., Gaussian), a rough estimate of $d_\lambda$ is sufficient to significantly reduce the bias. We will discuss this in detail in the final version of the paper.

**2. The assumption that** $\alpha \geq d$ (raised by Reviewers 2 and 4). As discussed in Remark 11, the assumption that $\alpha \geq d$ in Theorem 2 (and Table 1) is introduced *only* to account for the additional computational cost of the surrogate sketch when comparing to prior work (the cost is related to DPP sampling), but is not needed for the convergence rate. In fact, unlike in Determinantal Averaging [10], our convergence rate bound holds for any $\alpha \geq 1$, and it is by a factor of $d$ better than in [10] *for all* $\alpha$ (Table 1). It is worth pointing out that time complexity of DPP sampling is an ongoing area of research, and we believe that further improvements in this area will eliminate the need for the $\alpha \geq d$ assumption.

**3. Condition number dependence of the convergence rate result** (raised by Reviewers 3 and 4). Dependence on the condition number is standard in the analysis of distributed Newton-type methods, including GIANT [25], Determinantal Averaging [10], DANE [22], AIDE [21], DiSCO [26] and others. Our approach enjoys a merely logarithmic dependence of the number of iterations on the condition number (see Remark 3), which is at least as good as any of the listed methods. Following Reviewer 3's suggestion, we empirically compared the convergence rate of distributed Newton using the surrogate sketch (on the logistic regression task from Figure 3a) to a full unsketched gradient descent (GD) with backtracking line search, observing that GD required over 3x more iterations, compared to our method, until reaching $10^{-9}$ relative function value error. We will add these additional experiments to the final version.

**Reviewer 1**

- **Comparison of [25] and our work:** The main difference between [25] and our work is that our method gives unbiased estimates for the Newton step, which is not the case for [25]. This makes the most difference in the high $q$ (many workers) and high accuracy (small $\epsilon$) regime, which is important in massively parallel computing and federated learning. Also, from a practical point of view, simply applying rescaled regularization to the method of [25] (without using surrogate sketching) improves the error performance of [25] greatly, as illustrated in Figure 1.

**Reviewer 2**

- **Does a result analogous to Lemma 8 hold for other surrogate sketches?** Yes, we believe that a version of Lemma 8 can be shown for any surrogate sketch which admits a spectral approximation of $\mathbf{A}^\top \mathbf{A}$ (most of our proofs extend naturally to this general case, however some technical difficulties remain).

- **Advantage over using a single server with a preconditioner:** When we desire an accurate solution (i.e., small $\epsilon$), then the cost of performing $\log(1/\epsilon)$ iterations over all data can be very significant. By averaging the outputs of multiple workers, we can obtain better estimates for the Newton direction, and thus require far fewer than $\log(1/\epsilon)$ iterations over all data (see Remark 3).

- **Difference between this work and [9]:** While our proof techniques for showing the expectation formulas in Lemma 4 are indeed based on [9], the scaled regularization phenomenon is specific to our analysis ([9] did not observe this scaling). Furthermore, unlike us, [9] considers neither the algorithmic nor the measure concentration aspects of determinantal distributions. For example, they do not have an analogue of Lemma 8.

- **What is capital** $E$ **in Definition 2?** It is any event measurable with respect to the random (matrix) variable $\mathbf{X}$.

- **Is a surrogate sketch for a general random projection useful?** In terms of sampling from, say, a surrogate Gaussian sketch, we expect that this can be done efficiently (e.g., [12] shows how to sample from a slightly different determinant-rescaled Gaussian distribution). Furthermore, we believe that by bounding the discrepancy between a standard sketch and its surrogate, it will be possible to bound the bias of the standard sketch.

**Reviewer 3**

- **Linear/non-linear models:** We would like to clarify that the model is not necessarily linear but the update direction is obtained through solving a linear system. Sketched Newton's method works for convex cost functions in general, and our surrogate sketches apply there as well. For instance, Figure 7 in the appendix shows the error curves for distributed Newton sketch applied to a convex objective formulated using the log-barrier method.

**Reviewer 4**

- **Comparison of the least squares case and the general convex case:** The convergence rate in Theorem 2 (least squares case) is stated in terms of the $\ell_2$ norm *squared*, whereas Theorem 10 (general convex case) is stated in terms of the $\ell_2$ norm (not squared). Squaring the expression in Theorem 10 shows that the dependence on $\kappa$ is the same in both cases (also, it matches [10] and [25]). We will clarify this in the final version.

[Meta-Review · NeurIPS 2020]

The reviews all agreed that the paper contains interesting (even if not groundbreaking) ideas, and only expressed minor concerns and suggestions. Please take into account the updated reviews when preparing the final version to accommodate the requested changes.